

# The sensitivity of modeled snow accumulation and melt to precipitation phase methods across a climatic gradient

Keith S. Jennings[1,2,3,4] and Noah P. Molotch[1,2,5]

[1]Geography Department, University of Colorado Boulder, 260 UCB, Boulder, Colorado 80309, USA
[2]Institute of Arctic and Alpine Research, University of Colorado Boulder, 450 UCB, Boulder, CO 80309, USA
[3]Department of Geography, University of Nevada, Reno, 1664 N. Virginia Street, Reno, NV 89557, USA
[4]Desert Research Institute, 2215 Raggio Parkway, Reno, NV 89512, USA
[5]NASA Jet Propulsion Laboratory, 4800 Oak Grove Drive, Pasadena, CA 91109, USA

*Correspondence to*: Keith S. Jennings (keithj@unr.edu)

**Abstract.** A critical component of hydrologic modeling in cold and temperate regions is partitioning precipitation into snow and rain, yet little is known about how uncertainty in precipitation phase propagates into variability in simulated snow accumulation and melt. Given the wide variety of methods for distinguishing between snow and rain, it is imperative to evaluate the sensitivity of snowpack model output to precipitation phase determination methods, especially considering the potential of snow-to-rain shifts associated with climate warming to fundamentally change the hydrology of snow-dominated
areas. To address these needs we quantified the sensitivity of modeled snow accumulation and melt to rain-snow partitioning at research sites in the western United States. Simulations using the physics-based SNOWPACK model and 12 different precipitation phase methods indicated maritime sites were the most sensitive to method selection. Relative differences between the minimum and maximum annual snowfall fractions predicted by the different methods sometimes exceeded 100% at elevations less than 2000 m in the Oregon Cascades and California's Sierra Nevada mountains. This led to ranges in
annual peak snow water equivalent (SWE) typically greater than 200 mm, exceeding 400 mm in certain years. At the warmer sites, ranges in snowmelt timing predicted by the different methods were generally larger than 2 weeks, while ranges in snow cover duration approached 1 month and greater. Conversely, the three coldest sites in this work were relatively insensitive to the choice of a precipitation phase method with average ranges in annual snowfall fraction, peak SWE, snowmelt timing, and snow cover duration less than 18%, 62 mm, 10 d, and 15 d, respectively. Overall, sites with a greater proportion of
precipitation falling at air temperatures between 0°C and 4°C exhibited the greatest sensitivity to method selection. These findings have large implications for modeled snowpack water storage and land surface albedo at the warmer fringes of the seasonal snow zone.

## 1 Introduction

One of the most prominent impacts of climate warming has been a shift from snow to rain in temperate and cold
regions across the globe (e.g., Knowles et al., 2006; Trenberth, 2011), a trend that is expected to continue with further





increases in air temperature (Bintanja and Andry, 2017; Klos et al., 2014; O'Gorman, 2014; Safeeq et al., 2015). In order to assess how this change affects global hydroclimate, researchers have employed snow models, hydrologic models, and land surface models of varying degrees of complexity (e.g., Barnett et al., 2005). One trait many of these models share is the partitioning of rainfall and snowfall based on a spatially uniform air temperature threshold or a range between two thresholds

with a linear mix of liquid and solid precipitation in between. Recent work has called into question this simplistic treatment of precipitation phase (Feiccabrino et al., 2015; Harpold et al., 2017b) because of the pronounced spatial variability of rain-snow partitioning (Jennings et al., 2018b; Ye et al., 2013).

The use of a spatially uniform air temperature threshold is seemingly logical given the strong temperature-dependence of precipitation phase. Observational work has shown that precipitation is primarily solid at temperatures at and

below the freezing point (Auer Jr, 1974; Avanzi et al., 2014; United States Army Corps of Engineers, 1956) and that the probability of snowfall decreases following a sigmoidal curve as air temperature increases above 0°C (Dai, 2008; Fassnacht et al., 2013; Kienzle, 2008). However, the point at which the sigmoidal curve crosses 50% snow probability (i.e., the 50% rain-snow air temperature threshold) has been shown to vary significantly across the Northern Hemisphere (Jennings et al., 2018b). Thus, a single air temperature threshold, or range, cannot accurately represent precipitation phase partitioning across

large spatial extents (Raleigh and Lundquist, 2012). Part of this variability can be ascribed to relative humidity as recent work has shown snowfall is more probable at a given air temperature in more arid conditions (Froidurot et al., 2014; Gjertsen and Ødegaard, 2005; Jennings et al., 2018b). Surface air pressure also affects phase partitioning, but to a lesser degree than air temperature and humidity, with snowfall more common at higher air temperatures when surface pressure is lower (Ding et al., 2014; Jennings et al., 2018b; Rajagopal and Harpold, 2016).

Given the secondary controls exerted by humidity and surface pressure on the probability of rain versus snow, precipitation phase methods have been developed to leverage this information into more accurate rain and snow predictions. These methods include dew point temperature thresholds (Marks et al., 2013; Ye et al., 2013), wet/ice bulb temperature thresholds (Anderson, 1968; Harder and Pomeroy, 2013), and binary logistic regression equations that predict the probability of snow as a function of various meteorological quantities (Froidurot et al., 2014). In general, methods incorporating

humidity better predict precipitation phase than air temperature-only methods relative to observations across the Northern Hemisphere (Jennings et al., 2018b), likely due to their better representation of the hydrometeor energy balance (Harder and Pomeroy, 2013; Harpold et al., 2017b). Furthermore, the spatial variability of phase partitioning is reduced when using humidity information in addition to air temperature (Ye et al., 2013).

This wide variety of precipitation phase methods leads to variations in snowfall fraction—the percentage of annual

precipitation that falls as snow—approaching 30% or greater when applied to station meteorological data and reanalysis products (Harpold et al., 2017c; Jennings et al., 2018b; Raleigh et al., 2016). In general, warmer sites are more sensitive to precipitation phase method selection in terms of annual snowfall fraction variability, though it is less certain how this variability translates into divergences in simulated snow accumulation and melt. To that end, Harder and Pomeroy (2014) showed that precipitation phase method selection can produce ranges in annual peak SWE and snow cover duration of 160





mm and 36 d, respectively. However, this work only examined relatively cold research basins in Canada and did not consider the warmer mid-latitude, maritime climates that have been shown to be most "at risk" to the effects of climate warming on snow accumulation (e.g., Nolin and Daly, 2006). Similarly, other researchers have found higher air temperature thresholds generate greater annual peak SWE and increased snow accumulation during storm events at individual sites and basins

(Fassnacht and Soulis, 2002; Mizukami et al., 2013; Wayand et al., 2017; Wen et al., 2013).

We are therefore left with the question of how the sensitivity of modeled snow accumulation and melt to precipitation phase method selection varies across sites with different climatic characteristics. Considering over 1 billion people worldwide rely on mountain snowpacks for water resources (Barnett et al., 2005; Mankin et al., 2015), it is essential that models accurately represent precipitation phase partitioning as well as snowpack water storage and snowmelt timing.

Furthermore, snowpacks are highly reflective relative to bare ground, meaning simulated snow cover duration has a significant effect on modeled land surface albedo. These issues are further compounded when future warming-driven changes to snow accumulation and melt are taken into consideration, particularly if precipitation phase method selection induces uncertainty approaching that of the warming signal. Thus, it is necessary to quantify the baseline uncertainty in snow cover evolution due to the choice of a precipitation phase method, and then evaluate how the uncertainty relates to seasonal

climate at a diverse selection of sites.

The western United States offers a unique opportunity to perform such research for several reasons. One, the region includes both maritime and continental climates. Two, the region expresses a wide range of 50% rain-snow air temperature thresholds, increasing from ~1°C near the Pacific Coast to over 3°C in the Rocky Mountains (Jennings et al., 2018b). And three, model forcing and validation data are freely available through publicly funded networks. In the research presented

herein, we simulate eight years of snow cover evolution using 12 precipitation phase methods at sites that span a climatic gradient from warm maritime to cold continental with the goal of answering the following research questions:

1.   What is the sensitivity of annual snowfall fraction and modeled snow accumulation and melt due to precipitation phase method selection?

2.   How is the sensitivity controlled by air temperature, relative humidity, and precipitation?

## 2 Study sites and data

We selected sites in the western United States (Fig. 1) with long-term forcing and validation data that represented a range of snow conditions from transient snow with rain-on-snow and midwinter melt events to cold, deep seasonal snowpacks with little midwinter snowmelt. For this work, three stations at the HJ Andrews Experimental Forest were used to represent warm, maritime snowpacks. The two stations at the Southern Sierra Critical Zone Observatory (CZO) also have

warm, maritime climates, but seasonal snowpacks develop more consistently. The final maritime site is Dana Meadows in Yosemite National Park; however, this site consistently develops deep seasonal snowpacks due to considerably colder winter air temperatures than the other two maritime sites. The semi-arid Johnston Draw site forms part of the Reynolds Creek





Experimental Watershed and is located in the intermountain transition zone between maritime and continental climates. Finally, the two stations at Niwot Ridge are representative of cold continental locations. More information on the sites is presented in the text below and in Table 1.

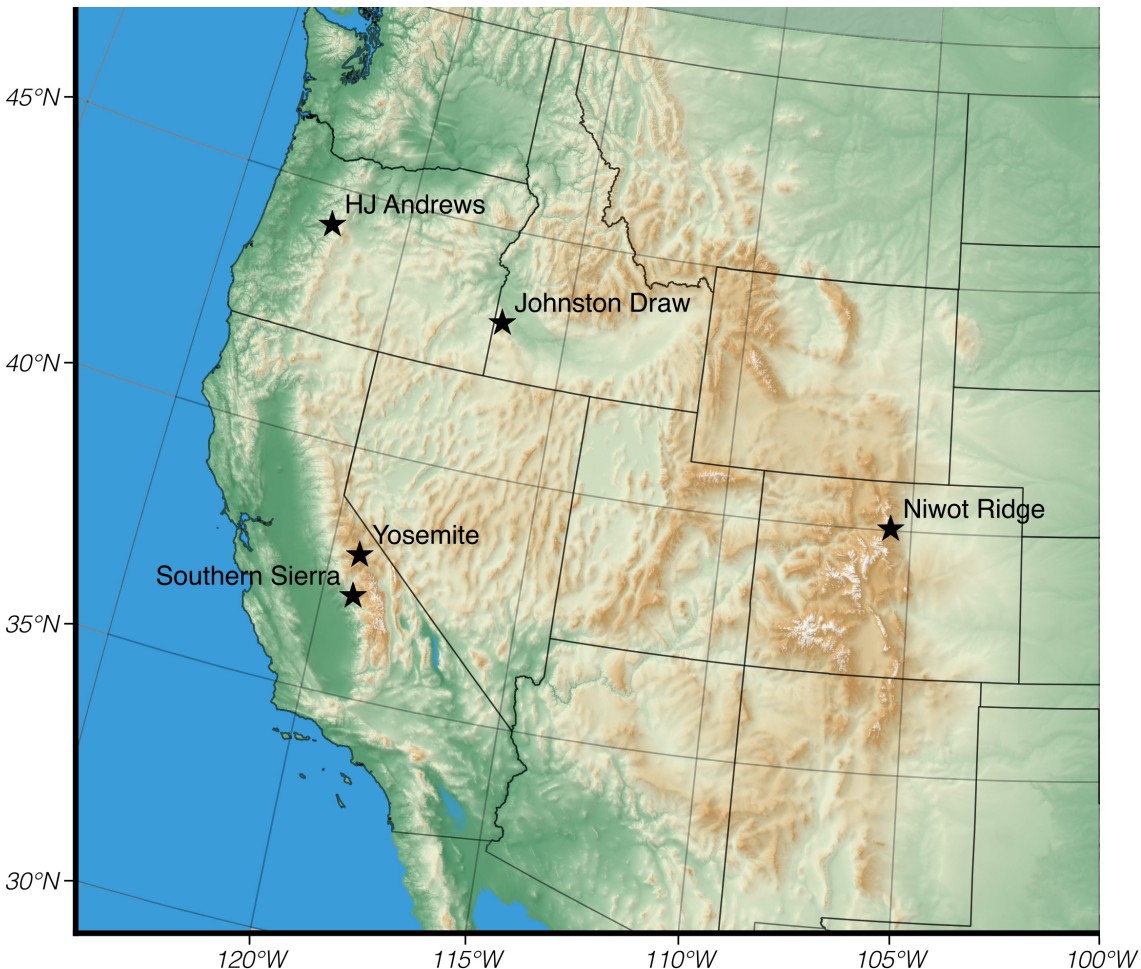

**Figure 1: The western United States showing the 5 study sites. Details on the stations at each site along with their meteorological characteristics are detailed in the following paragraphs and in Table 1.**

The HJ Andrews Experimental Forest (HJA), located in western Oregon, is part of the Long Term Ecological Research (LTER) network. We focused on the three meteorological stations with long-term forcing and validation data: Cenmet (HJA-CEN), Vanmet (HJA-VAN), and Uplmet (HJA-UPL). Due to its lower elevation, the HJA-CEN site only develops seasonal snowpacks during some winters, but is otherwise transient. HJA-VAN and HJA-UPL typically develop





seasonal snowpacks, but snow is transient in some years. Winter melt and rain-on-snow events are common throughout HJA (Harr, 1986; Jennings and Jones, 2015; Mazurkiewicz et al., 2008; Perkins and Jones, 2008). This site represents a typical maritime climate within the rain-snow transition zone.

The Upper (SSC-UPR) and Lower (SSC-LWR) Providence Creek stations in the Southern Sierra CZO (SSC) are
within the maritime zone of California's Sierra Nevada mountains and generally develop seasonal snowpacks. Reported annual snowfall fractions range between 20% and 60%, and rain-on-snow events can occur at both stations (Hunsaker et al., 2012). SSC-UPR and SSC-LWR can be either rain- or snow-dominated depending on the climate of a particular year (Hunsaker et al., 2012). This site represents maritime climates in the seasonal snow zone where winter melt events are frequent but snow cover persists throughout the winter.

The Dana Meadows station (YOS-DAN) is located within California's Yosemite National Park and is part of the Yosemite Hydroclimate Network (Lundquist et al., 2016). YOS-DAN receives significant winter precipitation, which produces snowpacks several meters deep due to cold winter temperatures (Lundquist et al., 2016; Rice et al., 2011). Although YOS-DAN has a maritime climate, annual snowfall fraction can exceed 90% (Lundquist et al., 2016) because of the station's high elevation and strongly seasonal precipitation. Winter melt makes up a relatively low proportion of annual
snowmelt at this elevation (Rice et al., 2011).

Johnston Draw (JD) is a sub-watershed within the larger Reynolds Creek Experimental Watershed, which is part of the CZO network in southwestern Idaho. Reynolds is within the rain-snow transition zone (Nayak et al., 2010) and has a semi-arid intermountain climate, bridging the divide between maritime and continental. We focused our simulations on three stations with co-located meteorological and snow depth measurements: 125 (JD-125), 124b (JD-124b), and 124 (JD-124).
Previous work has shown average annual snowfall fraction ranges from 39% at the lower station to 53% at the highest (Godsey et al., 2018). Similar to the HJA stations, seasonal snowpacks develop at the Johnston Draw stations in some years, but not in others. Due to high wind speeds and complex terrain, snow patterns vary across sites from year to year (Godsey et al., 2018). Additionally, winter melt and rain-on-snow events occur throughout the Reynolds Creek Experimental Watershed (Marks et al., 2001; Marks and Winstral, 2001).

The Niwot Ridge LTER (NWT) in Colorado's Rocky Mountains has a cold continental climate (Greenland, 1989) with previously reported annual snowfall fractions ranging between 63% and 80% (Caine, 1996; Knowles et al., 2015). The C1 station (NWT-C1) is in the subalpine area of NWT and Saddle (NWT-SDL) is situated above treeline in the alpine. Winter melt and rain-on-snow events are rare at both stations, particularly at NWT-SDL. High winter wind speeds are responsible for significant spatial variation in snow depth at NWT-SDL (Erickson et al., 2005; Litaor et al., 2008), while a
dense stand of lodgepole pine reduces the effect of wind on snow cover evolution at NWT-C1.





**Table 1. Station information plus average annual and December/January/February (DJF) climatic conditions ($T_a$ = air temperature, RH = relative humidity, VW = wind speed, and PPT = precipitation) for the 8 years of the study period (WY2004– WY2011).**

| Site | Station | Code | Elevation (m) | Annual $T_a$ (°C) | RH (%) | VW (m s$^{-1}$) | PPT (mm) | DJF $T_a$ (°C) | RH (%) | VW (m s$^{-1}$) | PPT (mm) | PPT* (%) |
|---|---|---|---|---|---|---|---|---|---|---|---|---|
| HJ Andrews | Cenmet | HJA-CEN | 1020 | 7.5 | 81.2 | 1.0 | 2308 | 1.7 | 86.3 | 1.0 | 957 | 41.5 |
| | Vanmet | HJA-VAN | 1275 | 7.0 | 76.8 | 1.2 | 2259 | 1.3 | 80.4 | 1.3 | 956 | 42.3 |
| | Uplmet | HJA-UPL | 1295 | 6.5 | 77.3 | 0.8 | 2841 | 0.7 | 81.6 | 0.8 | 1133 | 39.9 |
| Southern Sierra CZO | Lower Providence | SSC-LWR | 1753 | 8.4‡ | 68.3 | 0.9 | 1538 | 1.3 | 79.5 | 0.6 | 821 | 53.4 |
| | Upper Providence | SSC-UPR | 1981 | 9.1 | 57.4 | 1.2 | 1613 | 2.3 | 63.7 | 0.9 | 878 | 54.4 |
| Yosemite Nat. Park | Dana Meadows | YOS-DAN | 2987 | 1.4 | 55.6 | 1.3 | 811 | -5.5 | 62.7 | 1.4 | 468 | 57.7 |
| Johnston Draw | 125 | JD-125 | 1508 | 7.9 | 57.6 | 1.7 | 586 | -1.5 | 74.3 | 1.7 | 217 | 37.0 |
| | 124b | JD-124b | 1778 | 6.8 | 59.2 | 1.8 | 718 | -2.1 | 74.5 | 1.9 | 301 | 41.9 |
| | 124 | JD-124 | 1804 | 6.9 | 56.8 | 4.4 | 580 | -2.2 | 72.5 | 5.3 | 198 | 34.1 |
| Niwot Ridge | C1 | NWT-C1 | 3022 | 2.6 | 60.8 | 2.7 | 917 | -6.3 | 62.3 | 4.1 | 216 | 23.6 |
| | Saddle | NWT-SDL | 3528 | -0.7 | 64.3 | 8.5 | 1483 | -9.9 | 71.4 | 11.7 | 592 | 39.9† |

*Column corresponds to percentage of annual precipitation that falls during DJF.

‡Average $T_a$ values are cooler at SSC-LWR than SSC-UPR due to differences in vegetation and physiography at the two stations (M. Safeeq, personal communication, 20 June 2018). †High DJF precipitation percentage likely due to gage overcatch reduction factors. The alpine precipitation gage sees significant overcatch due to blowing snow (Williams et al., 1998) and reduction factors were developed relative to observed changes in the NWT-SDL snow pit SWE (Jennings et al., 2018a).

**3 Methods**

**3.1 Model setup and forcing data preparation**

We used the one-dimensional, physics-based SNOWPACK model (Bartelt and Lehning, 2002; Lehning et al., 2002a, 2002b) to evaluate the sensitivity of snow cover evolution to various precipitation phase methods. SNOWPACK is forced with air temperature ($T_a$), relative humidity (RH), wind speed (VW), incoming shortwave radiation (SW$_{in}$), incoming

longwave radiation (LW$_{in}$), and precipitation (PPT) at an hourly or longer time step. Part of our motivation for using SNOWPACK, in addition to the model's consistent performance in snow model studies (Etchevers et al., 2004; Rutter et al., 2009) and extensive validation (Jennings et al., 2018a; Lehning et al., 2001; Lundy et al., 2001; Meromy et al., 2015), was that it offers the user the option to include precipitation phase as part of the forcing data. In this scheme, the user can identify a time step as all-snow (0) or all-rain (1), or a mix of precipitation (decimal values between 0 and 1). Further details on the

precipitation phase methods implemented in this study are provided in Sect. 3.2 below and model validation is given in Appendix A.



We ran SNOWPACK at an hourly time step and kept model setup nearly identical across the sites in order to make the precipitation phase sensitivity results as comparable as possible. The only changes made to model setup were the meteorological measurement heights (Table S1), which were provided as part of the various forcing datasets. In some cases, this approach overlooked important changes to the snow accumulation and melt processes (e.g., snowfall interception,

enhancement of incoming longwave radiation) caused by forest cover, notably at the HJ Andrews site and, to a lesser extent, NWT-C1. However, we wanted the simulations to represent snow cover evolution without introducing the confounding hydrologic effects of interception and model representation thereof, meaning the canopy module for SNOWPACK was not activated at any of the sites. We acknowledge properly representing snow-forest interactions is critical to modeling snow in many basins (Lehning et al., 2006; Rutter et al., 2009) as tree cover exerts important controls on snow accumulation and melt

(Dickerson-Lange et al., 2017; Lundquist et al., 2013; Roth and Nolin, 2017). Future work should therefore examine how model representations of both vegetation and precipitation phase interact to produce uncertainty in modeled SWE.

Where possible, we relied on quality control and infilling methods from the dataset creators given their familiarity with meteorological processes at their respective sites. At HJA, the provided data were quality controlled, but not serially complete. We first infilled data with instruments at different heights located at the same station when those measurements

were available. We used linear regressions from the other stations to fill all other missing data. For the SSC stations, we performed an additional quality control routine based on Meek and Hatfield (1994) in order to clean up spurious data points. We then infilled missing data by regressing the two SSC stations. All other datasets were serially complete and we performed no further quality control or infilling procedures.

Additionally, none of the sites had $LW_{in}$ measurements available for the entirety of the study period. We used the

empirical estimates of $LW_{in}$ provided with the NWT and YOS-DAN datasets to force SNOWPACK. At the former, $LW_{in}$ was estimated as a function of $T_a$, RH, and $SW_{in}$ using the approaches of Angström (1915), Dilley and O'Brien (1998), and Crawford and Duchon (1999) as detailed in Jennings et al. (2018a). $LW_{in}$ was estimated at the latter (Lundquist et al., 2016) using the equations presented in Prata (1996) and Deardoff (1978). For the other sites, we used the empirical Unsworth and Monteith (1975) formulation that is included with the forcing data preprocessor MeteoIO (Bavay and Egger, 2014). At the

HJA stations, we bias-corrected the $LW_{in}$ estimate based on one year of $LW_{in}$ observations from HJA-VAN that showed a -56.9 W m$^{-2}$ wintertime bias. This was significantly larger in magnitude than the bias found in the Unsworth and Monteith (1975) estimate by Flerchinger et al. (2009), suggesting its performance is more spatially variable than previously noted. No bias corrections or additional methods were examined at the JD and SSC stations.

### 3.2 Precipitation phase methods

We evaluated a selection of precipitation phase methods found in the literature, including the more typical $T_a$ thresholds and ranges as well as methods incorporating humidity (Table 2). For the $T_a$, dew point ($T_d$), and wet bulb ($T_w$) thresholds, precipitation was designated as all-rain when the temperature was warmer than the threshold and all-snow when cooler than or equal to the threshold. When using the $T_a$ ranges, a linear mix of precipitation phase was given when $T_a$ fell





within the range during precipitation with all-rain above the warmer threshold and all-snow below the cooler threshold. The binary regression methods (Froidurot et al., 2014; Jennings et al., 2018b) computed the probability of snow ($p_{snow}$) as a function of $T_a$ and RH (Reg$_{Bi}$, Eq. 1) and as a function of $T_a$, RH, and surface pressure ($P_s$, Reg$_{Tri}$, Eq. 2). Precipitation was set to be all snow when $p_{snow} >= 0.5$ and rain when $p_{snow} < 0.5$:

$$p_{snow} = \frac{1}{1 + e^{(-10.04\ +\ 1.41T_a\ +\ 0.09RH)}} \tag{1}$$

$$p_{snow} = \frac{1}{1 + e^{(-12.8\ +\ 1.41T_a\ +\ 0.09RH+0.03P_s)}} \tag{2}$$

5  Each of the study sites included RH as part of their meteorological observations, but only the HJA and JD stations had observations of $T_d$, while no sites had long-term $T_w$ measurements. To keep precipitation phase methods constant across the sites, we calculated $T_d$ (Alduchov and Eskridge, 1996) and $T_w$ (Stull, 2011) as empirical functions of $T_a$ and RH. The empirical formulation tracked observed $T_d$ at JD with an $r^2$ of 1.0 and a slight cool bias of -0.3°C. There were no observations on which to validate the $T_w$ estimates, but Stull (2011) shows biases typically < 1.0°C.

10  It should be noted that although this work pursues a wide variety of precipitation phase methods, it is not wholly comprehensive. For example, some models fit a sigmoidal curve between two thresholds when assigning precipitation phase in a $T_a$ range (e.g., Fassnacht et al., 2013; Kienzle, 2008; Leavesley et al., 1995). However, we did not include this method because it should produce little variability in annual snowfall fraction relative to the linear $T_a$ ranges if a uniform distribution of air temperature and precipitation is assumed within the temperature range. Additionally, models of cloud microphysics are

15 increasingly used to simulate precipitation phase. The wide variety of microphysics schemes available suggests that a critical examination of these methods should be made, as well. However, such an analysis is beyond the scope of the current work.



**Table 2. Details on the precipitation phase methods used in this work. The temperature value for each threshold method is given in the "Rain-snow threshold" column. The "All-snow threshold" and "All-rain threshold" columns respectively give the $T_a$ values below which all precipitation is snow and above which all precipitation is rain for the $T_a$ range methods. The regression models compute phase as a function of meteorological conditions (Eqs. 1 and 2) during precipitation and are not associated with a threshold value. Due to a large variety of precipitation thresholds and ranges (Feiccabrino et al., 2015; Harpold et al., 2017b; Jennings et al., 2018b), the citations are listed if the values are approximate.**

| Category | Method | Rain-snow threshold (°C) | All-snow threshold (°C) | All-rain threshold (°C) | Citation(s) |
|---|---|---|---|---|---|
| $T_a$ threshold | $T_{a0}$ | 0.0 | NA | NA | (Jennings et al., 2018a; Lehning et al., 2002b*; Lynch-Stieglitz, 1994; Rajagopal and Harpold, 2016; Wen et al., 2013) |
| | $T_{a1}$ | 1.0 | NA | NA | |
| | $T_{a2}$ | 2.0 | NA | NA | |
| | $T_{a3}$ | 3.0 | NA | NA | |
| $T_a$ range | $T_{ar0}$ | NA | -0.5 | 0.5 | (Cherkauer et al., 2003; Tarboton and Luce, 1996; United States Army Corps of Engineers, 1956; Wayand et al., 2016; Wigmosta et al., 1994) |
| | $T_{ar1}$ | NA | -1.0 | 3.0 | |
| $T_d$ threshold | $T_{d0}$ | 0.0 | NA | NA | (Marks et al., 2013; Zhang et al., 2017) |
| | $T_{d1}$ | 1.0 | NA | NA | |
| $T_w$ threshold | $T_{w0}$ | 0.0 | NA | NA | (Anderson, 1968; Harder and Pomeroy, 2013; Marks et al., 2013) |
| | $T_{w1}$ | 1.0 | NA | NA | |
| Binary logistic regression | $Reg_{Bi}$ | NA | NA | NA | (Froidurot et al., 2014; Jennings et al., 2018b) |
| | $Reg_{Tri}$ | NA | NA | NA | |

*The SNOWPACK default is a 1.2°C $T_a$ threshold.

### 3.3 Evaluating the effect of precipitation phase method selection on snowfall fraction and simulated snow cover evolution

For water years (WY, 1 October of the previous calendar year to 30 September) 2004–2011, we simulated snowpack accumulation and melt at the 11 stations using the SNOWPACK model. Each station had a total of 12 unique model runs corresponding to the different precipitation phase methods. All forcing data and model setup remained the same across the runs at each site except for the precipitation phase method. For each site and for each of the different precipitation phase methods we quantified the average annual snowfall fraction, peak SWE magnitude, the timing of peak SWE, snowmelt rate, and snow cover duration (Fig. 2). For this work, snowmelt rate is computed as the daily average snowmelt rate between peak SWE timing and the first day where SWE = 0 mm. Snow cover duration is the total number of days when simulated SWE is greater than zero. For each of the sites we present the average simulated quantities noted above as well as the range and relative differences of snow metrics associated with the different precipitation phase methods. In this work, relative difference is defined as the percentage difference between the maximum and minimum snow metric value (e.g., if $T_{a0}$ produced a minimum peak SWE of 200 mm and $T_{a3}$ produced a maximum peak SWE of 400 mm, the relative difference would be 100%). Stations with greater variability in their snow cover evolution metrics were considered to be more sensitive to the choice of precipitation phase method.



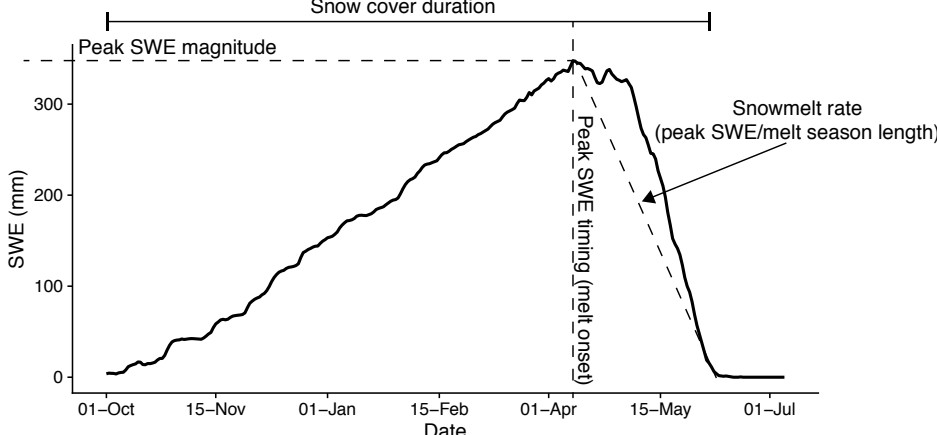

**Figure 2. Example niveograph showing seasonal snow cover evolution, adapted from Trujillo and Molotch (2014).**

### 3.4 Evaluating the relationships between climate and snow cover sensitivity

In addition to quantifying the variability introduced by the different precipitation phase methods, we evaluated the control exerted by daily meteorology and seasonal climate on snow cover evolution sensitivity at our study sites. We first examined how daily $T_a$ and RH introduced variability into simulated snowfall fraction. We did this by grouping all daily meteorological conditions in 1°C $T_a$ bins from -8°C to +8°C and 10% RH bins from 60% to 100% on days with precipitation. We then calculated the standard deviation in daily snowfall fraction within each bin across all sites and methods. Those results were used to determine the $T_a$ range that produced the greatest standard deviation in daily snowfall fraction. Next, we computed the proportion of December through May (Dec–May, i.e., winter and spring) precipitation that fell within that $T_a$ range at each site for each simulation year and used that percentage to predict annual snowfall fraction range with ordinary least squares regression. Finally, we quantified how Dec–May $T_a$ and PPT controlled variability in peak SWE at our study sites by computing a multiple linear regression with the two meteorological quantities acting as the predictor variables.

## 4 Results

### 4.1 Mean simulated snow cover properties

The study locations showed significant differences in simulated snow accumulation and melt. Values presented in Table 3 were computed by taking the mean and standard deviation of the given snow metric using all 12 simulations at each station, where each simulation corresponded to a different precipitation phase method. Mean peak SWE ranged from 73.1





mm at JD-124 to 1146.1 mm at HJA-UPL. The date of peak SWE, or melt onset, also displayed large variability with values ranging from 24 January at JD-125 to 13 May at NWT-SDL. Melt rates were all greater than 10 mm d$^{-1}$ during the ablation season except for the JD stations and the greatest melt rates were simulated at HJA-UPL and NWT-SDL. Snow cover duration was greatest at NWT-SDL at 241.1 d, while snow cover was simulated for less than 3 months, on average, at JD-125 and JD-124.

**Table 3. Mean snow cover evolution metrics for the 11 stations. Each mean and standard deviation was calculated across all water years and all precipitation phase methods.**

| Station | Peak SWE (mm) | | Peak SWE date | | Melt rate (mm d$^{-1}$) | | SCD (d) | |
|---|---|---|---|---|---|---|---|---|
| | Mean | SD | Mean | SD (d) | Mean | SD | Mean | SD |
| HJA-CEN | 522.7 | 252.9 | 16-Feb | 22.0 | 15 | 3.9 | 158.4 | 28.2 |
| HJA-VAN | 643.1 | 305.9 | 14-Feb | 22.2 | 14.5 | 3.2 | 173.1 | 27.9 |
| HJA-UPL | 1146.1 | 469.9 | 14-Mar | 23.0 | 24.9 | 7.3 | 201.1 | 22.4 |
| SSC-LWR | 531.9 | 160.1 | 8-Mar | 19.0 | 17.6 | 3.6 | 145.6 | 27.8 |
| SSC-UPR | 617.9 | 298.8 | 5-Mar | 26.6 | 17.6 | 6.0 | 149.2 | 35.6 |
| YOS-DAN | 674.4 | 236.7 | 18-Mar | 17.5 | 10.9 | 4.1 | 208.2 | 40.3 |
| JD-125 | 83.4 | 46.5 | 24-Jan | 28.5 | 4 | 1.5 | 78.1 | 31.5 |
| JD-124b | 177.5 | 87.6 | 1-Feb | 25.8 | 5.7 | 2.5 | 122.4 | 23.9 |
| JD-124 | 73.1 | 35.0 | 2-Feb | 31.4 | 3.5 | 2.8 | 77.6 | 30.7 |
| NWT-C1 | 407.2 | 78.5 | 22-Apr | 10.8 | 11.9 | 2.8 | 225.3 | 19.2 |
| NWT-SDL | 915 | 234.2 | 13-May | 10.0 | 24.4 | 10.1 | 241.1 | 14.9 |

### 4.2 Effect of precipitation phase method on simulated snowfall fraction

Average annual snowfall fraction (all methods, all years) ranged from 32.3% at the HJA-CEN station to 92.4% at the YOS-DAN station (Table 4, Fig. 3). In this case, more strongly seasonal precipitation at YOS-DAN (Table 1) produced a higher annual snowfall fraction than NWT-SDL, despite the former station's warmer average $T_a$. These two stations also had the lowest ranges at 10.1% and 10.3%, respectively, suggesting precipitation phase method selection was less important relative to the other stations. Conversely, the range in annual snowfall fraction simulated by the different methods was greater than 18% at all remaining stations, reaching a maximum of 32.3% at SSC-LWR. For all sites except YOS and NWT, relative differences were greater than 30%. In some years at HJA, SSC, and JD, the relative difference between the minimum annual snowfall fraction and the maximum exceeded 100%, meaning the methods producing the most snow simulated more than double the annual snowfall fraction of those producing the most rainfall. The greatest relative difference in annual snowfall fraction of 126.9% was simulated at HJA-CEN, more than 10-times greater than at YOS-DAN and NWT-SDL.





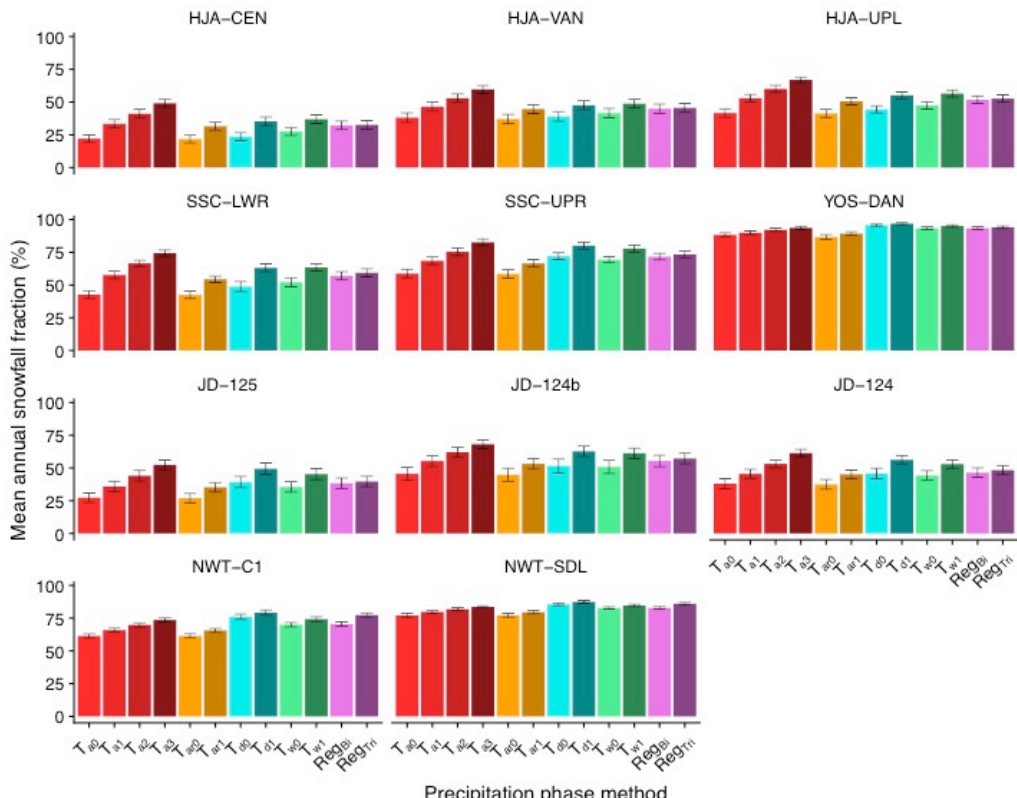

**Figure 3. Mean annual snowfall fraction at the 11 study stations for the 12 different precipitation phase methods. The whiskers represent the standard error of annual snowfall fraction for the 8 simulation years. For this plot and all subsequent figures showing the station data, the maritime sites are shown in the top two rows, the intermountain site is in the third row, and the continental site is in the bottom row.**





**Table 4. Statistics for average annual snowfall fraction computed using the 12 different precipitation phase methods across all simulations years at the 11 study stations. The range was calculated by subtracting the lowest average annual snowfall fraction from the highest average annual snowfall fraction at each site. The relative difference was then computed as the range divided by the minimum and multiplied by 100%. $T_{a0}$ and $T_{ar0}$ typically produced the lowest average annual snowfall fractions, while $T_{a3}$ and $T_{d1}$ led to the highest average annual snowfall fractions.**

| | Annual snowfall fraction (%) | | |
| Station | Average | Range | Relative difference |
| --- | --- | --- | --- |
| HJA-CEN | 32.3 | 27.4 | 126.9 |
| HJA-VAN | 45.5 | 22.6 | 61.1 |
| HJA-UPL | 51.8 | 25.7 | 62.3 |
| SSC-LWR | 56.8 | 32.3 | 76.7 |
| SSC-UPR | 71.2 | 25.0 | 42.8 |
| YOS-DAN | 92.4 | 10.1 | 11.6 |
| JD-125 | 39.1 | 26.0 | 97.1 |
| JD-124b | 55.7 | 23.2 | 51.8 |
| JD-124 | 47.9 | 23.9 | 63.6 |
| NWT-C1 | 70.4 | 18.2 | 29.7 |
| NWT-SDL | 82.4 | 10.3 | 13.4 |

### 4.3 Effect of precipitation phase method on simulated snow accumulation and melt

There were marked differences between the stations in terms of the effect precipitation phase method choice had on seasonal snow cover evolution. Figure 4 presents the simulated mean daily SWE of all simulation years at the study stations along with the difference between the minimum and maximum mean daily SWE produced by the precipitation phase methods. At HJA, SSC, and JD, differences increased throughout the accumulation period, reaching a maximum after peak SWE during the snowmelt season. At NWT and YOS, the differences were typically negligible throughout the accumulation season as cold winter and early spring temperatures produced little divergence in the amount of snowfall versus rainfall simulated by the different methods. At these stations, differences in the mean daily SWE produced by the precipitation phase methods did not appear until approximately the date of peak SWE. Mean daily SWE differences were always less than 90 mm at NWT and YOS, while they sometimes exceeded 200 mm at HJA and SSC. Differences were typically small in magnitude at JD, but were proportionally large due to low mean daily SWE values.





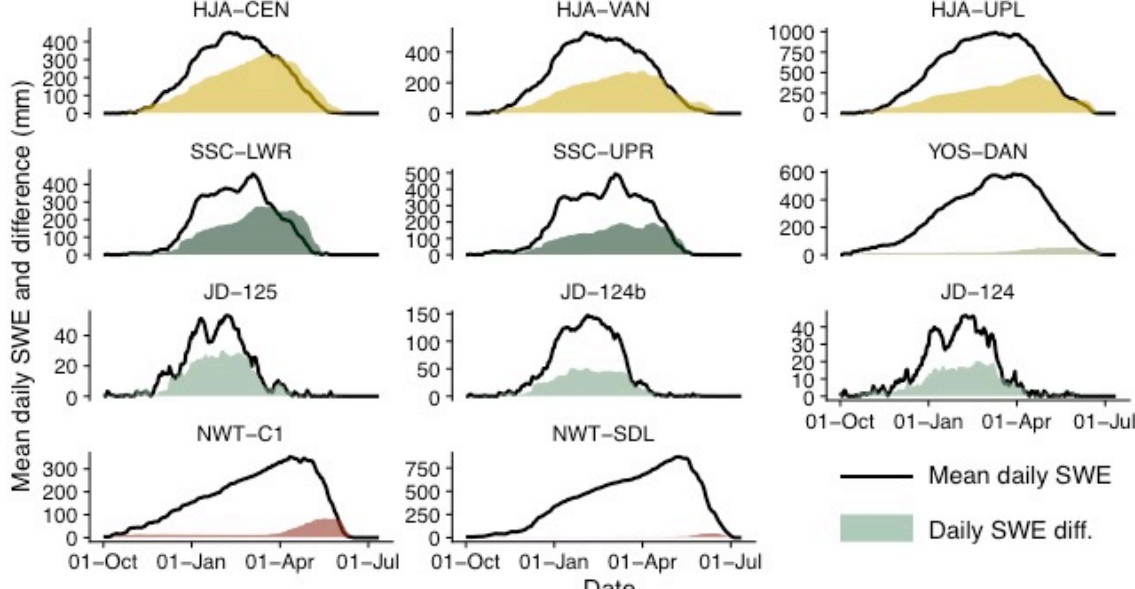

**Figure 4. Mean daily SWE (solid black line) and the difference between maximum and minimum mean daily SWE (shading) at the study stations. The mean daily SWE was computed by averaging the simulated SWE on each day for all precipitation phase methods across the simulation years. The difference was calculated by subtracting the minimum mean daily SWE from the maximum mean daily SWE produced by the different precipitation phase methods (mean daily SWE plots broken out by precipitation phase method can be viewed in Figs. S1–S11). The $T_{a0}$ and $T_{ar0}$ methods typically produced the minimum mean daily SWE, while $T_{a3}$ and $T_{d1}$ produced the maximum.**

Breaking down the analysis to the individual snow cover evolution metrics reveals more differences in the sensitivity of the sites to precipitation phase method selection (Fig. 5). In terms of peak SWE range, the HJA and SSC stations were most sensitive, with average ranges all greater than 200 mm, exceeding 400 mm in some years (Fig 5a). Conversely, YOS and NWT were relatively insensitive as their average ranges were all less than 65 mm. The largest annual range in peak SWE at the NWT and YOS stations was just 90.8 mm at NWT-C1, which was considerably less than the maximum peak SWE range of 592.5 mm simulated at HJA-UPL. Although the JD stations showed little sensitivity in terms of range with average annual peak SWE differences less than 55 mm, they expressed significant sensitivity when looking at relative differences (Table 5) due to their low mean annual peak SWE (Table 3). Thus, percentage-wise, JD was as sensitive as the two warm maritime sites to the selection of a precipitation phase method. At JD, HJA, and SSC it was common for the relative difference between minimum and maximum modeled peak SWE to be well above 50%, meaning a significant proportion of water was simulated to have run off using one precipitation phase method versus being stored in the snowpack using another method. This is in stark contrast to the 4.0% and 1.8% relative differences at YOS-DAN and NWT-SDL.



JD and HJA were also sensitive to precipitation phase method selection in terms of peak SWE date (Fig. 5b) with 4 of the 6 stations having average ranges greater than 2 weeks. In some simulation years, peak SWE date ranges exceeded 1 month at HJA, JD, and SSC. We found the greatest differences in peak SWE dates were generally simulated on years with low/transient snow cover. In these cases, late-season precipitation was simulated as rain by the low $T_a$ thresholds and snow

by the high $T_a$ thresholds, meaning an early SWE maximum was recorded as the peak in the former case and a late SWE maximum in the latter case. Compared to the other stations, peak SWE date ranges were generally small at NWT-SDL and YOS-DAN with an average range of just 0.8 d at the former and 2.5 d at the latter.

Similar sensitivities were simulated for snow cover duration (Fig. 5c) with the warm maritime sites and JD being the most impacted by precipitation phase method choice. JD-125 had the greatest average range in annual snow cover

duration at 42.0 d and all other ranges at JD and HJA were greater than 26.8 d. SSC-LWR and SSC-UPR expressed slightly lower average ranges at 20.9 d and 18.1 d, respectively. NWT-C1 approached the sensitivity of the warmer stations, while NWT-SDL and YOS-DAN were again the least sensitive. Relative differences were greatest at JD (Table 5) because simulated snow cover was typically of a shorter duration compared to the other sites (Table 3). The average relative difference at JD-125 of 120.4% meant that snow cover simulated using the $T_{a3}$ threshold lasted twice as long as snow cover

using the $T_{a0}$ threshold. Notably, there was an order of magnitude of difference between JD, HJA, and SSC and YOS and NWT with average relative differences in snow cover duration greater than 10% at the former three sites and less than 10% at the latter two.

Finally, differences among the stations were relatively low for melt rate (Fig. 5d). Ranges for the different stations were relatively similar to one another with the interquartile ranges generally showing some degree of overlap. JD stations

had the greatest sensitivity in terms of relative differences (Table 5) due to their low mean annual melt rates, which were an order of magnitude lower than those simulated at the other sites (Table 3). Overall, melt rate at YOS-DAN was the least affected by precipitation phase method selection in terms of range and relative difference. It should be noted here again that the forcing data were kept constant for the different modeling scenarios—only the precipitation phase methods were varied. Thus, any changes to melt rate were caused by shifts in snowmelt timing and by the hydrologic and energy balance impacts

of rain versus snow.





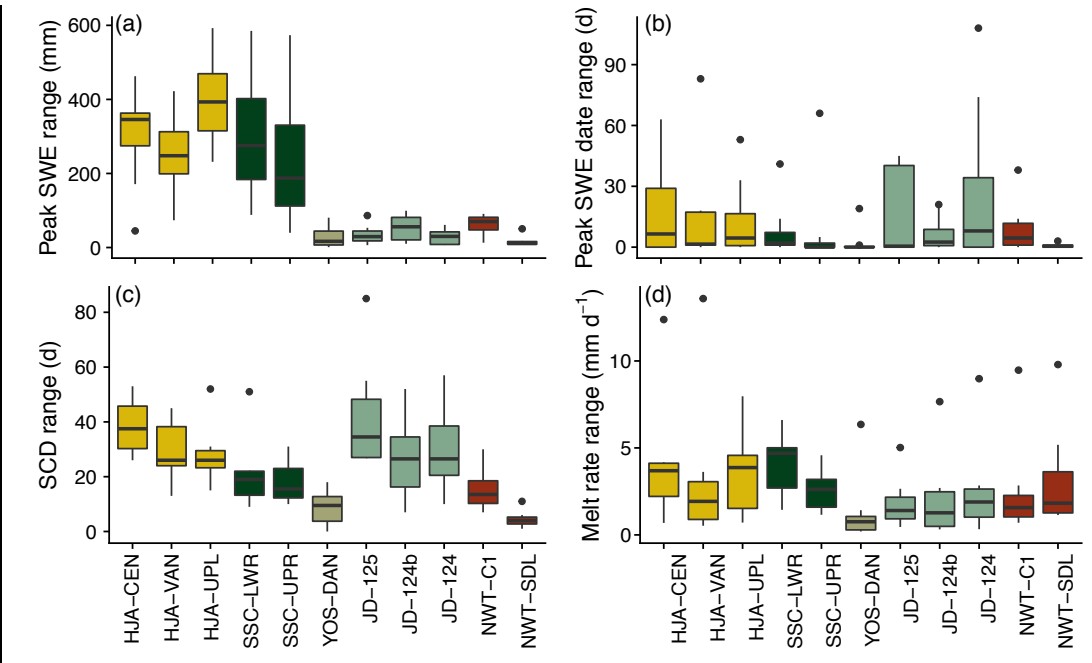

**Figure 5.** The annual range in simulated peak SWE (a), peak SWE date (b), snow cover duration (c), and melt rate (d) due to precipitation phase method selection at the study stations.

**Table 5.** Average relative differences in annual peak SWE, snow cover duration, and melt rate at the 11 stations. Relative differences were not computed because for peak SWE date because the relative difference value would change depending on if day of year or day of water year were used in the calculation.

| | Average relative difference (%) | | |
| --- | --- | --- | --- |
| **Station** | **Peak SWE** | **SCD** | **Melt rate** |
| HJA-CEN | 86.6 | 28.0 | 33.2 |
| HJA-VAN | 55.2 | 19.6 | 27.5 |
| HJA-UPL | 49.1 | 15.4 | 19.6 |
| SSC-LWR | 78.6 | 14.9 | 26.7 |
| SSC-UPR | 43.9 | 13.4 | 15.6 |
| YOS-DAN | 4.0 | 4.8 | 11.5 |
| JD-125 | 74.6 | 120.4 | 220.2 |
| JD-124b | 54.7 | 28.7 | 47.8 |
| JD-124 | 71.9 | 72.4 | 235.5 |
| NWT-C1 | 16.9 | 7.0 | 26.0 |
| NWT-SDL | 1.8 | 1.9 | 13.0 |





### 4.4 Climatic controls on precipitation phase method sensitivity

In general, daily snowfall fraction standard deviation was greatest at daily $T_a$ values between 0°C and 4°C (Fig. 6a). RH provided a secondary control, with greater daily snowfall fraction variability at lower RH values (Fig. 6b). Overall, the largest standard deviations in snowfall fraction were simulated at daily RH less than 80% and $T_a$ between 1°C and 3°C.

5    However, it should be noted that 75.2% of all precipitation recorded at the study stations occurred in the 90%–100% RH bin. Therefore, although daily snowfall fraction standard deviations were highest at lower RH values, the majority of the variability in snowfall fraction was an effect of $T_a$. In this context, the percentage of Dec–May precipitation that fell within the 0°C–4°C $T_a$ range explained 80.1% of the variance in annual snowfall fraction standard deviation across the study sites (Fig. 7).

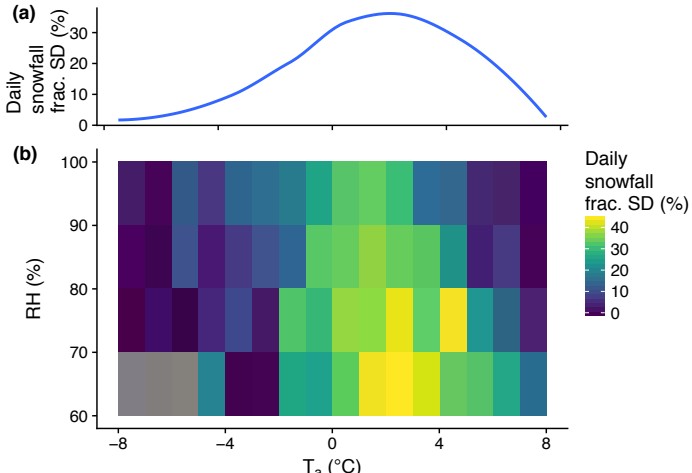

**Figure 6. The standard deviation of daily snowfall fraction as a function of $T_a$ (a) and as a function of $T_a$ and RH (b). We binned the meteorological quantities within the ranges shown and calculated the standard deviation of snowfall fraction per $T_a$ bin (a) and $T_a$/RH bin (b) using simulated precipitation phase from all stations and all methods.**





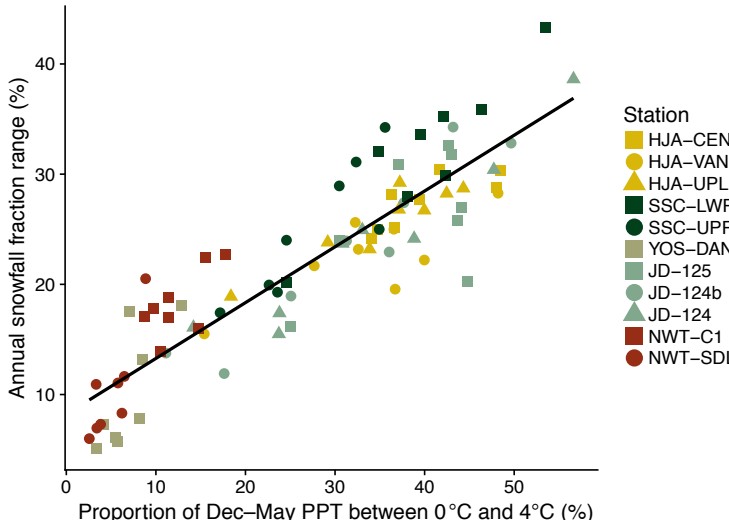

**Figure 7.** Range in annual snowfall fraction as predicted by the proportion of Dec–May PPT falling between 0°C and 4°C. Each point represents one simulation year at a station identified by the color and shape. The black line of best fit was calculated using ordinary least squares regression ($r^2 = 0.80$, *p-value* < 0.0001).

We next evaluated how sensitivity in peak SWE was related to seasonal climate. A multiple linear regression with Dec–May $T_a$ and Dec–May PPT as the predictor variables explained 80.6% of the variance in the range of annual peak SWE at the stations (Fig. 8). In this case, warmer $T_a$ and increased PPT were both associated with greater ranges in the peak SWE simulated by the different precipitation phase methods. This meant the maritime sites HJA and SSC had the greatest sensitivity to precipitation phase method due to their relatively warm $T_a$ and high PPT values. Conversely, moderate PPT

values and lower $T_a$ led to minimal sensitivity at the cold continental NWT stations and the cold maritime YOS-DAN station. Again, the effect of $T_a$ on sensitivity was manifest in the data. In high snowfall years at NWT-SDL, Dec–May PPT approached that of the low Dec–May PPT years at HJA and SSC. However, despite the increased PPT at NWT-SDL, the range in peak SWE predicted by the different precipitation phase methods remained low. Additionally, the multiple linear regression performed here is likely only valid for the range of climatic conditions at our study sites. For example,

extrapolating the regression to $T_a$ values above 5°C would indicate greater peak SWE sensitivity for a given PPT value. However, moving towards increasingly warmer $T_a$ would likely lead to lower peak SWE ranges due to the increasing probability of rainfall versus snowfall.





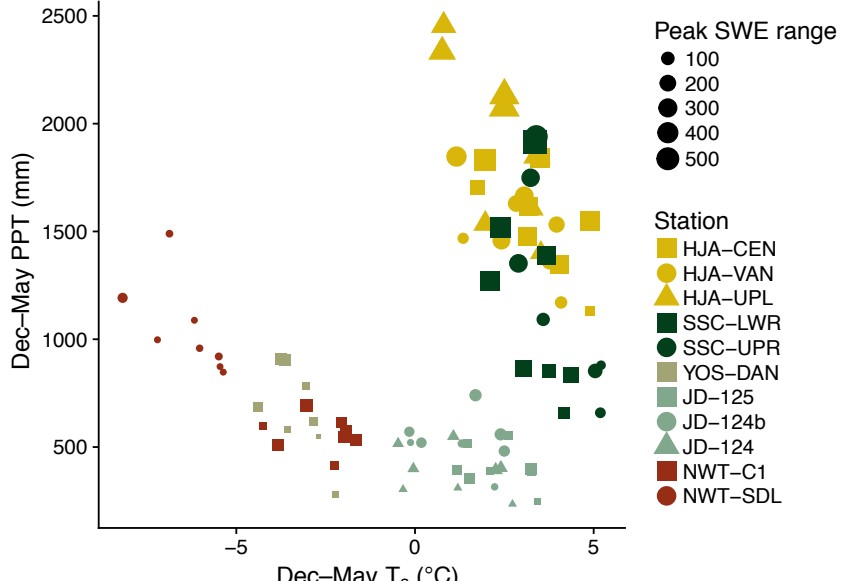

**Figure 8. Range in annual peak SWE as simulated by the different precipitation phase methods at the 11 study stations. Each point represents one simulation year at a given station and larger points correspond to larger differences in maximum minus minimum peak SWE. Predicting the peak SWE range as a function of Dec–May $T_a$ and PPT using multiple linear regression**
**yielded an $r^2$ of 0.81 (*p-value* < 0.0001).**

## 5. Discussion

### 5.1 A best precipitation phase method?

In this work we showed that the selection of a precipitation phase method introduces variability in modeled snow accumulation and melt. The different methods also expressed variable performance relative to observations of SWE and
10 snow depth (Fig. A1). In terms of mean bias, the binary regression models, $Reg_{Bi}$ and $Reg_{Tri}$, as well as the $T_{a1}$ threshold provided the best performance with average values between 3.1 mm and 9.1 mm compared to observed SWE and between -1.7 mm and 6.9 mm compared to observed snow depth. Conversely, the $T_{a0}$, $T_{a2}$, $T_{a3}$ thresholds and the $T_{ar0}$ range provided the worst performance with $T_{a2}$ and $T_{a3}$ overpredicting snow accumulation and $T_{a0}$ and $T_{ar0}$ underpredicting snow accumulation by upwards of 100 mm relative to observed SWE and 200 mm and greater relative to observed snow depth.
There was relatively little divergence in $r^2$ values across the methods, with only a 0.07 and 0.08 difference between the maximum and minimum average $r^2$ values for SWE and snow depth, respectively. The lowest $r^2$ values were produced by the $T_{a0}$ threshold and $T_{ar0}$ range, while $T_{d1}$, $T_{w1}$ and the higher $T_a$ thresholds produced the highest values.



Previous work has shown that, in general, methods incorporating humidity information outperform $T_a$-only methods (Harder and Pomeroy, 2013; Jennings et al., 2018b; Marks et al., 2013; Ye et al., 2013). In that context, one can consider the $Reg_{Bi}$ model as a baseline given its top rank in a Northern Hemisphere precipitation phase method comparison (Jennings et al., 2018b). Our study showed that $Reg_{Bi}$ typically produced low biases relative observed snow depth and SWE (Fig. A1) and

led to snow cover evolution metrics that were neither extremely high nor low relative to the other methods examined in this work. The $T_{a2}$, $T_{a3}$, $T_{d1}$ and $T_{w1}$ thresholds produced greater peak SWE and longer snow cover duration, while the lower thresholds led to less snow accumulation and shorter snow cover duration. Additionally, our model-based study showed that uncertainty in daily snowfall fraction peaked at $T_a$ between 0°C and 4°C (Fig. 4.8), which is the same $T_a$ range reported by Ding et al. (2014) in which precipitation phase methods exhibit degraded performance relative to observations.

Despite the analyses presented in this work, it is important to note that uncertainties in forcing data, model structure and parameters, as well as a lack of precipitation phase observations prevent this research from being a referendum on the "best" precipitation phase method for snow modeling. Our aim was to quantify how snow simulations were affected by the choice of precipitation phase method across a climatic gradient. We did not create optimized model setups at each site, but rather kept model setup consistent in order to compare the sensitivity of phase partitioning without introducing other

uncertainties. Thus, the low $r^2$ and higher bias values at HJA-VAN, NWT-SDL, and JD-124 (Fig. A2) could likely be improved with model tuning, but we did not pursue such an approach.

**5.2 Assumptions and limitations**

Snow modeling studies are hindered by inherent uncertainties in model structure (Essery et al., 2013; Etchevers et al., 2004; Rutter et al., 2009; Slater et al., 2001) and forcing data (Lapo et al., 2015; Raleigh et al., 2015, 2016). While the

research presented herein shows that precipitation phase method should be considered another critical component of model uncertainty, our work was also likely affected by the aforementioned issues in structure and forcing data which can be seen in the variability of model performance at the different sites (Fig. A2). In this work, we used the well-validated, physics-based SNOWPACK model, but past research has shown there is no best snow model and that model performance varies both within and across study sites (e.g., Rutter et al., 2009). Therefore, our use of a single model may overestimate or

underestimate the sensitivity of snow cover evolution to precipitation phase method at certain sites and points in time. Future research should therefore focus on how model choice affects the sensitivity of simulated snow cover evolution to precipitation phase method.

In addition to the uncertainties introduced by the SNOWPACK model, we used empirical methods to estimate $T_d$ and $T_w$, which could affect rain-snow partitioning. We were satisfied with the performance of the $T_d$ method as it strongly

matched $T_d$ observations from Johnston Draw (Sect. 3.2). However, there were no observations of $T_w$ on which to validate the Stull (2011) method, which was optimized for standard surface pressure and for a range of $T_a$ and RH values. The figures in Stull (2011) show that pressure-induced uncertainty in $T_w$ is generally less than 1°C when RH > 50%. Additionally, the total percentage of precipitation observations falling within the Stull (2011) $T_a$ and RH ranges was between 94.3% and 100%





at our stations. Thus, we expect only marginal uncertainty to be introduced by the empirical methods. However, precipitation phase and hydrometeor temperature are strongly related to $T_w$ (Harder and Pomeroy, 2013), suggesting there should be enhanced monitoring of $T_w$ at research sites.

Furthermore, our research only examined methods that partition precipitation phase using surface meteorological
quantities such as $T_a$ and RH. Atmospheric and climate models can also be used for hydroclimatic simulations either through direct coupling in earth systems models or as forcing data for land surface models. Many such models employ microphysics schemes to assign and track precipitation phase from the formation of a hydrometeor, through various atmospheric layers, to the land surface. For example, the Weather Research and Forecasting (WRF) model (Skamarock et al., 2005) has been used to simulate snow cover accumulation and ablation over large study domains in the western United States when coupled to a
land surface model (Ikeda et al., 2010; Musselman et al., 2017a; Rasmussen et al., 2011). WRF has also been used to model the elevation of the rain-snow transition line in order to evaluate which basin areas are receiving solid or liquid precipitation during storm events (Minder et al., 2011). In addition, work from the 5[th] phase of Coupled Model Intercomparison Project (CMIP5) has shown that climate models produce different snowfall fractions due to variations in both climate and precipitation phase method (Krasting et al., 2013). In CMIP5, some models utilize microphysics schemes, while others
assign precipitation phase at the land surface using methods similar to the ones presented in this work. Therefore, understanding and quantifying the sensitivity of model output due to precipitation phase method selection is important for both hydrologic and climate modeling studies.

**5.3 Physical mechanisms controlling sensitivity to phase method**

The warm maritime sites HJA and SSC expressed the largest peak SWE ranges from precipitation phase method
selection (Fig. 5). These ranges were typically larger than 200 mm and sometimes exceeded 400 mm with relative differences usually greater than 50%, indicating large uncertainty in snowpack water storage. Additionally, peak SWE date ranges typically exceeded 2 weeks at these stations, meaning the timing of snowmelt onset was also affected by precipitation phase method. These large variations in snow cover evolution were likely due to the combined effect of reduced frozen mass entering the snowpack and subsequent changes to the snowpack energy balance. For the former, both HJA and SSC had high
proportions of precipitation falling between 0°C and 4°C (Fig. 7), which led to wide ranges in annual snowfall fraction (Table 4). The methods producing lower annual snowfall fractions (e.g., $T_{a0}$ and $T_{ar0}$) generally corresponded to reduced snow cover duration (Fig. 6) simply because there was less frozen mass to melt. In other words, the energy required to melt the entire snowpack was reduced relative to the methods producing higher snowfall fractions, and the snowpack could be melted over a shorter time period.

Compounding the response of the warm maritime sites was the fact that snow and rain have different fates when they enter a snowpack with resultant effects on the snowpack energy budget. Snowfall can increase snowpack cold content (Jennings et al., 2018a), refresh surface albedo (Clow et al., 2016; Molotch et al., 2004; Molotch and Bales, 2006; Painter et al., 2012; United States Army Corps of Engineers, 1956), and provide dry pore space that must reach field capacity with





liquid water before runoff can begin (Bengtsson, 1982; Seligman et al., 2014). Rainfall, conversely, can advect heat to the snowpack (Marks et al., 1998), infiltrate and run off (Harr, 1981, 1986), or be refrozen in the snowpack if there is cold content to be satisfied. In this context, the precipitation phase methods that produced more rainfall affected snow cover evolution not just through reduced frozen mass but also through changes to the snowpack energy budget.

**5.4 Why precipitation phase matters to climate warming simulations**

The shift from snow to rain in cold and temperate regions across the globe is expected to continue with further warming. Future air temperature increases will likely produce reduced snowfall fractions (Klos et al., 2014; Lute et al., 2015; Safeeq et al., 2015), lower peak SWE values (Adam et al., 2009), earlier snowmelt onset (Stewart et al., 2004), slower snowmelt rates (Musselman et al., 2017a), and changes to the intensity and location of rain-on-snow events (Musselman et al., 2018). These warming-driven changes will impact both water resources availability (Barnett et al., 2008) and land surface albedo (Déry and Brown, 2007). Most "at risk" to reductions in snowfall fraction and snow accumulation are areas with winter $T_a$ near 0°C (Nolin and Daly, 2006). Concerningly, our work shows it is precisely these areas that have the greatest modeled snow cover accumulation and melt sensitivity to precipitation phase method selection.

Harpold et al. (2017c) showed that future changes to snowfall fraction are moderated or exacerbated by the choice of a precipitation phase method, depending on the area's relative humidity. However, how this uncertainty affects the conclusion of climate change predictions is typically not discussed. In the context of the work presented herein, there should be a focus applied to areas where the baseline variability in peak SWE, snowmelt onset, and snow cover duration due to precipitation phase method approaches or exceeds the simulated change in the associated snowpack properties with warming. In warm maritime climates, research has shown peak SWE may decrease by upwards of several hundred millimeters as warming continues (e.g., Cooper et al., 2016; Leung et al., 2004; Minder, 2010; Musselman et al., 2017b), which is near the range of peak SWE sensitivity values reported in this work. Precipitation phase method selection is also likely to impact simulations of future warm snow droughts where anomalously warm winters are associated with low peak SWE (Harpold et al., 2017b). In addition, snow cover duration variability due to precipitation phase method selection in earth systems models may affect simulations of the snow-albedo feedback, which is the amplification of surface warming due to reduced snow cover (Hall, 2004; Hall and Qu, 2006). As climate warming shifts new areas towards the winter and spring average $T_a$ values (0°C–4°C) that lead to the greatest uncertainty in rain-snow partitioning, our research suggests that uncertainty in future hydroclimatic states will be exacerbated by precipitation phase method selection.

**6 Conclusion**

In this work we simulated seasonal snow cover evolution using the SNOWPACK model forced with 12 different precipitation phase methods at 11 study stations spanning a climatic gradient from warm maritime to cold continental. We found the choice of a precipitation phase method introduced significant variability into simulated snow accumulation and




melt. Warm maritime sites were the most sensitive to method selection with relative differences in annual snowfall fraction near and above 100% and ranges in peak SWE typically greater than 200 mm, exceeding 400 mm in certain years. At these sites the different methods produced ranges in snowmelt timing and snow cover duration that were generally longer than 2 and 3 weeks, respectively. Conversely, the YOS-DAN and NWT-SDL stations exhibited the lowest sensitivity to

precipitation phase method selection with relative differences in annual snowfall fraction between 11.6% and 13.4%. Peak SWE ranges were typically less than 30 mm for these two stations, while average snowmelt onset date ranges were only 0.8 d and 2.5 d at YOS-DAN and NWT-SDL, respectively.

The spatially variable sensitivity of snow cover evolution was primarily a result of climatic differences between the stations. Increased Dec–May $T_a$ and PPT were associated with greater peak SWE ranges across the different precipitation

phase methods. This meant the maritime sites HJA and SSC, with significant winter and spring PPT, were most affected by precipitation phase method selection. Overall, we found stations with a high proportion of Dec–May PPT falling at $T_a$ between 0°C and 4°C to be more sensitive than those with less PPT in that $T_a$ range. This is troublesome considering climate warming is expected to push new areas in the seasonal snow zone towards winter $T_a$ near 0°C and above. It is therefore critical that future work examine the relationship between the effect of warming on snow cover evolution and the model

variability that results from precipitation phase partitioning uncertainty, particularly in areas undergoing a snow-to-rain transition.

**Data availability**

Forcing and validation data can be accessed at the following sites (as of 2019-02-11):

- HJ Andrews LTER: http://dx.doi.org/10.6073/pasta/c96875918bb9c86d330a457bf4295cd9 and
http://andlter.forestry.oregonstate.edu/data/ (for sub-daily data)
- Southern Sierra CZO: https://eng.ucmerced.edu/snsjho/files/MHWG/Field/Southern_Sierra_CZO_KREW (Hunsaker et al., 2012)
- Johnston Draw (Reynolds Creek CZO): https://doi.org/10.15482/USDA.ADC/1402076 (Godsey et al., 2018)
- Yosemite Dana Meadows: http://hdl.handle.net/1773/35957 (Lundquist et al., 2016)
- Niwot Ridge LTER: https://doi.org/10.6073/pasta/1538ccf520d89c7a11c2c489d973b232 (Jennings et al., 2018a) and https://doi.org/10.6073/pasta/f62b0a3741737c871958cf7e63c089e0 (Williams, 2016)

**Appendix A: Model validation**

As noted in the Methods section, model setup was kept constant at all the sites, no parameter tuning was performed, and the SNOWPACK canopy module was not activated. This was done to minimize the introduction of confounding factors

and to keep the simulation results as comparable as possible. Figure A1 displays the mean bias and $r^2$ values for the different



precipitation phase methods relative to observations of SWE and snow depth, aggregated across all the stations. Differences in method performance are discussed in depth in Section 5.1.

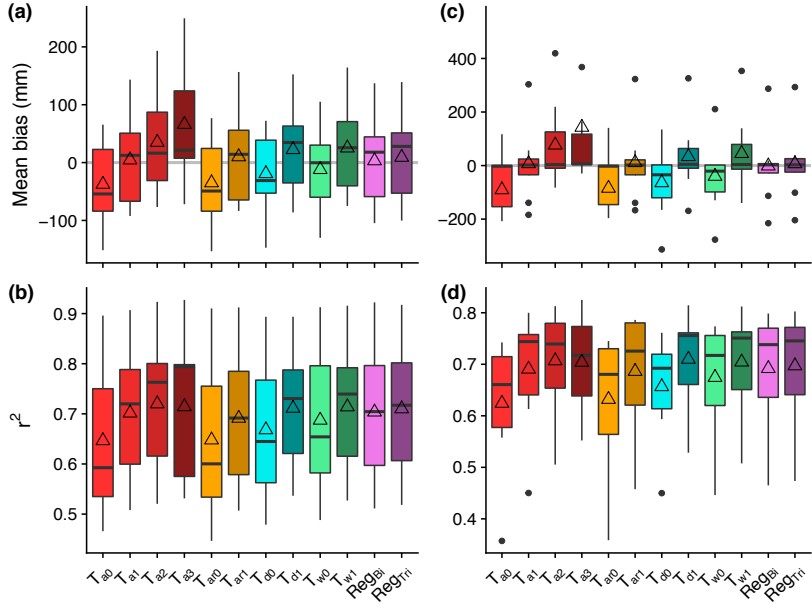

**Figure A1. Mean bias (top row) and r$^2$ (bottom row) values for the SNOWPACK simulations relative to observed SWE (a,b) and**
5    **snow depth (c,d). The boxplots show the median, interquartile range, minimum, maximum, and outlying values for each objective function for the different precipitation phase methods at all stations. The open triangles indicate the mean objective function value for that precipitation phase methods at all stations.**

Figure A2 presents model performance relative to observed SWE and snow depth at the different sites. Mean biases were lowest at the NWT stations and at SSC-UPR relative to SWE observations and at the JD stations and SSC-LWR
10   relative to snow depth observations. Average r$^2$ values were between 0.65 and 0.91 for SWE except at NWT-SDL (0.52) and HJA-VAN (0.51), and 0.61 and 0.79 for snow depth except at JD-124 (0.46). The variability of model output at the different stations is representative of both inconsistent snow model performance (Etchevers et al., 2004; Rutter et al., 2009) and the difficulty of accounting for wind processes in point snow models (Raleigh et al., 2015).





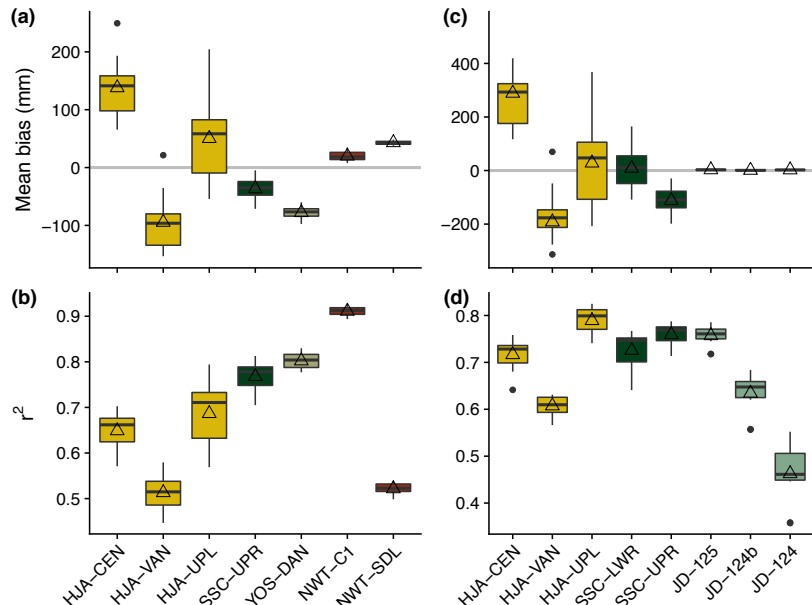

**Figure A2. Mean bias (top row) and r² (bottom row) values for the SNOWPACK simulations relative to observed SWE (a,b) and snow depth (c,d). The boxplots show the median, interquartile range, minimum, maximum, and outlying values for each objective function for the different precipitation phase methods at a given station. The open triangles indicate the mean objective function value for all precipitation phase methods at that station.**

### Author contributions

KSJ and NPM designed the study. KSJ performed the analyses and wrote the manuscript. NPM provided feedback and edited the manuscript.

### Acknowledgments

KSJ was supported by a NASA Earth and Space Science Fellowship (16-EARTH16F-378). The LTER and CZO networks are funded by the United States National Science Foundation. We offer our thanks to Don Henshaw (HJ Andrews LTER) and Mohammed Safeeq (Southern Sierra CZO) for their assistance with the respective datasets. We are also grateful for the other dataset providers for making their data freely and easily accessible.





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
