# Peer review of "The sensitivity of modeled snow accumulation and melt to precipitation phase methods across a climatic gradient"

_Hydrology and Earth System Sciences, 2019_

## Referee Comment (RC1) · Bettina Schaefli (Referee) · 25 Apr 2019

This well written paper analyzes a key question for snow hydrology, which is the impact of precipitation phase algorithms on snow water equivalent (SWE) modelling in different climates. The paper studies four more or less different methods of precipitation phase computation (each with different portioning parameters) and assesses the impact of the methods on different snow accumulation and melt metrics, obtained with the model SNOWPACK at five different locations in the US. The methods are based on temperature thresholds and on bilinear regression. The analysis gives an answer to

the general question of how important it is to carefully choose the precipitation phase method for different climates.

A drawback of the study is that it is purely simulation-based and does not use observed SWE data to push the study further. In fact, with the observed SWE data and SNOW-PACK, it might have been possible to estimate actual daily or hourly snow accumulation amounts and compute best parameter values for the studied precipitation phase methods at the selected stations. This way, it would have been possible to judge how critical deviations from these best estimates would be at the different sites. In other words, this would allow to answer questions like "how critical is it to have a 1°C error in the air temperature threshold at a warm site as opposed to a cold site"? "How important is it to use dew point or wetbulb temperature at warm sites versus at cold sites?"

This having said, the study is nevertheless worth publishing and interesting for the readers of HESS. Below some general and detail comments.

**General comments**

I would not say that a study tests 12 different methods if only a few methods are tested with different parameter values; this oversells the study in the abstract. I would in fact say that the study tested four different methods: based on air temperature (with different 50% thresholds and different transition ranges, some of the ranges being 0), based on dew point and wet bulb temperature and based on binary regression.

A key analysis of the paper is the one of "Climatic controls on precipitation phase method sensitivity".(section 4.4); it analyzes how the results vary with air temperature. Air temperature sensitivity is, however, built into each method in a different way. In the case of daily snowfall fraction: the fact that it shows the highest standard deviation for air temperatures between 0 and 4 C simply expresses the fact that several methods use thresholds in this range. The result would look different if the thresholds were between -2 and 2 C. This should be better reflected in the discussion of of the results.

[Figure]

In general, the conclusion that precipitation falling in the range 0 – 4 C explains much of the variation observed across the methods comes from the choice of the threshold values. Without actual comparison to observed data, the results are hard to generalize. Why is there no comparison to actual SWE-derived thresholds?

Furthermore, when reading the results section where actual SWE curves are presented for the first time, it is a little disappointing to see that all studied sites show a typical seasonal snow cover with significant accumulation over many weeks. The most sensitive sites would typically be the ones where the snow cover might build up several times during the winter.

**Detailed comments**

- The abstract does not mentioned what types of methods have been tested nor whether they have been compared to reference data or which method performed best

- Introduction: it would have been interesting to shortly discuss how /where precipitation phase is actually observed; as far as I am aware of, actual precipitation phase observations are crucially missing at most places.

- Introduction: the manuscript focuses its discussion on snow-hydrological models. How do meteorological forecast models determine the limit (elevation) of snow fall? Completing the literature review with this respect would complete the picture

- P. 2: "In general, warmer sites are more sensitive to precipitation phase method selection in terms of annual snowfall fraction variability, though it is less certain how this variability translates into divergences in simulated snow accumulation and melt. " This statement is given without reference. In what is the apparently previously known result different from your own findings?

- Study sites: It might be useful to know the variability of the daily air temperature around the seasonal mean (ie. the anomalies, obtained e.g. by fitting a sine curve to air temperature as in the work of Woods, 2009. It is this variability that will tell something about the probability of switching from accumulation to melting conditions and about a site sensivitiy to the chosen temperature threshold.

- Methods: it is not clear at this stage that all stations always show a seasonal snow cover (significant accumulation over several weeks), which is important for the concept of "peak SWE" to be meaningful

- the current definition of snowmelt rate is probably over sensitive to spurious shifts from a primary to a secondary SWE peak, which could reduce the melt duration sensibly; how could this measure be made more robust? Similar comment applies to the peak SWE date that is discussed in the results section. Is this measure useful? Minor modifications of SWE accumulation can switch the SWE peak date between a spurious primary or secondary peak (Figure 4 suggest that stations with two peaks might exist, but I might be mistaken).

- P. 14 "meaning a significant proportion of water was simulated to have run off using one precipitation phase method versus being stored in the snowpack". This not well formulated since rainfall does not necessarily run off. It can infiltrate and recharge the groundwater.

- Section 4.4: Here, standard deviations are calculated across the results of all 12 computation methods. Standard deviation does not seem to be a good measure to quantify the variability of values that do not come from an actual sample of a given process but of values pertaining to different methods. (Besides: how are standard deviations obtained? First per method and then averaged over all methods?)

Woods, R.A., 2009. Analytical model of seasonal climate impacts on snow hydrology: Continuous snowpacks. Advances in Water Resources, 32(10): 1465-1481. DOI:10.1016/j.advwatres.2009.06.011

---

## Referee Comment (RC2) · Jonathan Conway (Referee) · 30 Apr 2019

Review of paper hess-2019-82 "**The sensitivity of modeled snow accumulation and melt to precipitation phase methods across a climatic gradient**" by Keith S. Jennings and Noah P. Molotch

Jono Conway

This paper presents a systematic evaluation of the impact of precipitation phase partitioning on modelled snowfall and snowpack evolution. Multi-year datasets from 11 stations across 5 locations in the western United States are used to drive simulations with a sophisticated snowpack physics model. The effect of parameter and algorithm choice are assessed for a range of commonly used parameterisations. The authors relate the modelled sensitivity to average climate characteristics. Snowfall and maximum accumulated snow in warmer maritime locations with high precipitation and winter temperatures between 0 and 4 degrees Celsius are found to be most sensitive to precipitation partitioning, while snowfall in colder inland locations are found to be less sensitive.

The manuscript is well written and with good figures and a clear systematic structure. It addresses a topic of high interest and relevance internationally. However, there are some areas that should be addressed before the paper could be accepted for publication.

**Major comments**

While the paper is framed as a comparison of methods used to partition precipitation, the results mainly reflect the range of Ta thresholds used (0 to 3 C) rather than the choice of parameterisation. This is in part due to the use the range metric on results that are generally are bounded by the two extreme Ta thresholds. The abstract and conclusions should reflect this (i.e. being explicit about choice of parameter values and/or parameterisation rather than using the ambiguous term "method". If the authors wish to make general statements, then using "precipitation partitioning" would be more appropriate. It is well established the Ta alone is a poor predictor of precipitation phase, so to really compare methods, those that perform poorly against observed SWE (e.g. Ta0 and Ta3) should be removed from the analysis. This would highlight the differences induced by using different parameterisations that have a sound physical basis. If the Ta0 and Ta3 options are to be retained, then further justification for their inclusion should be given in the methods section. The dependence of the results (especially the range of Ta with a large range in modelled snow) on the range of Ta thresholds used should also be discussed. Perhaps the use of a standard deviation or similar metric rather than a range metric would put the focus on the choice of parameterisation. Further specific comments address this issue.

Timing and magnitude of SWE ranges seem mainly related to snowfall and accumulation, whereas as range of melt rate does not have high sensitivity or clear relation to climate. This should be clearer in the abstract and conclusions.

The abstract and conclusions need to highlight the novel aspects of the results presented here and provide clearer recommendations for future research. While the analysis is comprehensive, the result is not entirely new and, in my opinion, there are other results in the paper that could (and should) be highlighted in addition to the main result that the relative differences are largest in maritime snowpack. For example, the fact that using threshold or ranges for Ta (for the same 50% crossover) do not produce large differences in the snowpack, or that partitioning choice has little effect on snowmelt rate and the effects are dominated by snowfall. At present, the authors recommendations for future researchers are unclear.

The use of multiple linear regression is probably not appropriate here, but if retained should be presented and discussed more fully.

**Specific comments (page-line)**

1-15 please be clear the study modelled non-vegetated snowpacks only.

4-3 Given that they form a key part of the results, please include average values for Tw and Td in Table 1.

7-26 The large bias in LWin is concerning – perhaps the influence of vegetation on the measurements whereas LWin is modelled for non-vegetated location? This should be discussed when presenting the validation results in Figure A2.

17-8 "80.1% of the variance in annual snowfall fraction standard deviation" – the figure caption and methods describes this as the "range in annual snowfall fraction" – please clarify which it is and correct.

18-1 Figure 6 and 7 – given that the range in snowfall fraction is driven primarily by the two extreme air temperature threshold methods (Ta0 and Ta3) these results are presumably quite sensitive to the choice of the Ta thresholds? Please discuss and if possible show the sensitivity of the results to the choice of threshold.

18-6 Looking at the figure, it seems that a multiple linear regression may not be appropriate. There seems to be two groupings – highly sensitive warm and wet locations, less sensitive drier locations that span both warm and cold locations. Also, given that the equation is not presented nor used further, and the issues discussed with extrapolating the equations, please consider removing the regression. If it is retained, please present the equation and display contours of predicted values on Figure 8 so that the reader can visualise the predicted relationships.

19-10 The validation results presented in the appendix should be included in the results or methods section, especially as they form part of the discussion, rather than simply an intermediate methodological step.

20-3 "In that context, one can consider the RegBi model as a baseline given its top rank in a Northern Hemisphere precipitation phase method comparison". Please describe and discuss the results presented here (figure A1) that seem to show similar performance for a range of methods that incorporate humidity information. The discussion as it is not balanced and does not accurately reflect the results presented. Please revise.

20-11 "a referendum." This does not seem an appropriate term – please revise. You could either give an expert view based on the results presented here, or cite others work.

20-24 "Therefore, our use of a single model may overestimate or underestimate the sensitivity of snow cover evolution to precipitation phase method at certain sites and points in time." This statement is very broad  - more effort is needed to quantify and discuss the uncertainty of the model simulations.

21-23 "These large variations in snow cover evolution were likely due to the combined effect of reduced frozen mass entering the snowpack and subsequent changes to the snowpack energy balance". More detailed results are needed to support this statement. For example, the change in snowfall mass and albedo could be shown to illustrate the importance of the direct and indirect effects on snowpack mass balance.

22-3 "In this context, the precipitation phase methods that produced more rainfall affected snow cover evolution not just through reduced frozen mass but also through changes to the snowpack energy budget." These results are not shown here (they could be?) so this statement is speculation. Please revise.

22-25 "winter and spring average Ta values (0°C–4°C) that lead to the greatest uncertainty in rain-snow partitioning," I would argue that the uncertainty is not in the actual rain-snow partitioning, but rather due to the use of an inappropriate parameterisation (only Ta) which requires a wide range of parameter tuning. Please revise.

23-1 Please mention that no clear relationship was found for snowmelt rate in the conclusions – this is still a key result and an important caveat to the earlier statement that "precipitation phase method introduced significant variability into simulated snow accumulation and melt".

23-30 How was the r2 calculated here? the average r2 of hourly SWE/snowdepth or something else? Please include in the text and figure caption.

24-1 Given the poor performance of some methods (Ta0 ,Ta3, Tr0) should they be excluded from the analysis? If not, further discussion is needed.

24-6 "at all stations." Given that SWE and snowdepth are only presented for some sites in Figure A2, I presume not all sites contribute to averages here? Please list the sites that contribute to each of the SWE and snowdepth validation statistics in the text or caption.

Figure A2 – why is the snowdepth bias 0 for the JD sites?

**Editorial comments:**

10-6 "daily Ta and RH" do you mean "daily average Ta and RH"?

16-5 "not computed because for" -> "not computed for"

---

## Author Comment (AC1) · 31 May 2019

We thank Dr. Schaefli for their thoughtful review. Our responses can be found in blue throughout the following text. Please note, tables and figures specific to this response document are given with the prefix R (for example, Table R1 in the comment below). Tables and figures in the manuscript are referred to by numbers only.

This well written paper analyzes a key question for snow hydrology, which is the impact of precipitation phase algorithms on snow water equivalent (SWE) modelling in different climates. The paper studies four more or less different methods of precipita- tion phase computation (each with different portioning parameters) and assesses the impact of the methods on different snow accumulation and melt metrics, obtained with the model SNOWPACK at five different locations in the US. The methods are based on temperature thresholds and on bilinear regression. The analysis gives an answer to the general question of how important it is to carefully choose the precipitation phase method for different climates.

A drawback of the study is that it is purely simulation-based and does not use observed SWE data to push the study further. In fact, with the observed SWE data and SNOW-PACK, it might have been possible to estimate actual daily or hourly snow accumulation amounts and compute best parameter values for the studied precipitation phase meth- ods at the selected stations. This way, it would have been possible to judge how critical deviations from these best estimates would be at the different sites. In other words, this would allow to answer questions like "how critical is it to have a $1^{\circ}$C error in the air temperature threshold at a warm site as opposed to a cold site"? "How important is it to use dew point or wetbulb temperature at warm sites versus at cold sites?"

We agree this is a drawback of both this study and many other snow modeling research projects. There are, unfortunately, scant direct observations of precipitation phase in mountain regions. One of the few studies we are aware of that uses observations of precipitation phase—in this case snow board measurements—showed rain-snow partitioning errors can lead to significant biases in modeled snow accumulation at a site with a maritime climate similar to the HJ Andrews and Southern Sierra (Wayand et al., 2017). There is also evidence suggesting an optimized air temperature threshold varies throughout the snow season (Storck et al., 2002), meaning no single air temperature threshold (or range) would be applicable across sites and times.

At our study sites, there are no direct observations of precipitation phase, but we were interested in pursuing your question further. Table R1 below shows the optimized rain-snow air temperature threshold using four different data sources for each station. The second and third columns (Map and Obs.) correspond to data from earlier work that

examined the spatial variability of rain-snow partitioning across the Northern Hemisphere (Jennings et al., 2018). The methods, quoted from the paper, are as follows:

> *"To construct a spatially continuous 50% rain–snow $T_s$ [air temperature] threshold product across the Northern Hemisphere, we applied the optimized bivariate model to the MERRA-2 gridded reanalysis dataset[63,64]. Hourly 2 m $T_s$, specific humidity (q), $P_s$, and precipitation data were accessed from 1980 through 2007 and summarized to a daily time step. RH was calculated from the MERRA-2 data using an empirical equation as a function of q, $P_s$, and $T_s$. Daily snowfall probability was then simulated for each grid cell using the bivariate model when precipitation was greater than 1 mm and $T_s$ fell within the range of −8 to 8 °C. We then calculated the 50% rain–snow $T_s$ threshold by fitting the hyperbolic tangent to binned estimates of snowfall frequency per MERRA-2 grid cell using Eq. 1."*

> *"We classified precipitation reports as either rain or snow using the World Meteorological Organization precipitation phase categories described in detail in Dai[40,61]. Precipitation amounts were not included in the dataset and we removed sleet as well as potential mixed-phase observations from the analysis because the relative proportions of solid and liquid precipitation during such events were not reported (i.e., it was impossible to quantify the amount of precipitation falling as snow versus rain). The classification of precipitation events was then used to quantify the rain–snow frequency per 1 °C $T_s$ bin from –8 to 8 °C at each station. In other words, if there were 100 total precipitation observations from 1 to 2 °C, 75 of which were snow, the snowfall frequency in that bin would be 75.0%. We then calculated the 50% rain–snow $T_s$ threshold for each station using the approach of Dai[40], where a sigmoidal curve is fit to observations of snowfall frequency per 1 °C $T_s$ bin from –8 to 8 °C using a hyperbolic tangent function:*

$$T_{50} = \frac{\tanh^{-1}\left(\frac{F}{a} + d\right)}{b} + c \tag{1}$$

> *where $T_{50}$ equals the 50% rain–snow $T_s$ threshold (°C), F equals snowfall frequency (in this case 0.5, dimensionless), and a, b, c, and d are the fitting parameters (dimensionless)."*

The fourth and fifth columns in Table R1 use changes in SWE and snow depth to estimate a rain-snow air temperature threshold. We used a modified version of the approach of Rajagopal and Harpold (2016) to predict precipitation phase by designating a

daily increase of SWE or snow depth as snowfall and a zero change or decrease as rainfall when precipitation was greater than 2.54 mm and SWE or snow depth was greater than 0 mm. As with the Map and Obs. methods detailed above, we then binned snowfall frequency per 1°C air temperature bin (Figures R1 and R2) and computed the rain-snow air temperature threshold using Eq. 1 above. The SWE approach yielded values that approximated the Map and Obs. methods, but the depth-derived values were significantly lower. We would thus argue that this method was not appropriate for our purposes, although previous work has shown it to reasonably estimate precipitation phase at subdaily time scales (e.g., Marks et al., 2013; Zhang et al., 2017).

**Table R1.** Optimized rain-snow air temperature thresholds for each station in the study using four different data sources: 1-Map) The spatially continuous threshold map from Jennings et al. (2018) created using reanalysis data from MERRA-2 and the bivariate binary logistic regression model; 2-Obs.) The observed threshold from the closest meteorological station (Jennings et al., 2018); 3-SWE) The threshold inferred from changes in SWE at each study station (Fig. R1); 4-Depth) The threshold inferred from changes in snow depth at each study station (Fig. R2). An NA indicates there were insufficient data to estimate the threshold from SWE and/or snow depth.

| | Optimized rain-snow air temperature threshold (°C) | | | |
| --- | --- | --- | --- | --- |
| Station | Map | Obs. | SWE | Depth |
| HJA-CEN | 1.19 | 1.12 | 1.29 | -0.24 |
| HJA-VAN | 1.19 | 1.12 | 0.8 | -0.84 |
| HJA-UPL | 1.19 | 1.12 | -0.4 | -0.81 |
| SSC-LWR | 1.7 | 1 | NA | 0.14 |
| SSC-UPR | 1.7 | 1 | 0.87 | -0.34 |
| YOS-DAN | 2.21 | 2.78 | NA | NA |
| JD-125 | 2.25 | 1.25 | NA | -0.97 |
| JD-124b | 2.25 | 1.25 | NA | -1.91 |
| JD-124 | 2.25 | 1.25 | NA | 0.41 |
| NWT-C1 | 2.84 | 2.34 | 3.57 | NA |
| NWT-SDL | 2.84 | 2.34 | NA | NA |

[Figure]

**Figure R1. Snowfall frequency per 1°C air temperature bin as computed from SWE data. On days with precipitation > 2.54 mm, an increase in SWE was designated as a snowfall event, while a zero change or decrease in SWE was designated as rainfall.**

[Figure]

**Figure R2. Snowfall frequency per 1°C air temperature bin as computed from snow depth data. On days with precipitation > 2.54 mm, an increase in snow depth was designated as a snowfall event, while a zero change or decrease in snow depth was designated as rainfall.**

Returning to the question of "how critical is it to have a 1∘C error in the air temperature threshold at a warm site as opposed to a cold site," we analyzed the effect of deviating by 1°C from the mean threshold as calculated from the Map and Obs. columns in Table R1. In this context we rounded to the nearest integer degree to be consistent with our thresholds, giving the HJA stations a 1°C threshold, SSC a 1°C threshold, YOS a 2°C threshold, JD a 2°C threshold, and NWT a 3°C threshold. Because we did not include a 4°C air temperature threshold in our phase methods, we could only analyze a negative deviation at NWT. In Table R2 below, we present the mean peak SWE, peak SWE day of water year (DOWY), and snow cover duration (SCD) using the optimized air temperature threshold (center column, abbreviated Thresh.), the optimized threshold minus 1°C (left column, Thresh - 1°C), and the optimized threshold + 1°C (right column, Thresh + 1°C). Consistent with our findings in the paper, the warm maritime HJA and SSC stations are profoundly affected by deviations from the optimized threshold. Differences at these sites produced by deviating by ±1°C from the optimized thresholds range between 141 and 403 mm for peak SWE, 1 and 16 d for peak SWE DOWY, and 9 and 29 d for SCD. Compare this to 1 to 10 mm for peak SWE, 0 to 1 d for peak SWE DOWY, and 1 to 5 d for SCD at the YOS and NWT stations. The consistent story is again that threshold choice makes a much larger impact at a warm site relative to a cold one.

**Table R2.** Mean peak SWE, peak SWE DOWY, and SCD at the study stations using an optimized air temperature threshold as well as -1°C and +1°C deviations from the threshold.

| Station | Mean peak SWE (mm) | | | Mean peak SWE DOWY (d) | | | Mean SCD (d) | | |
|---|---|---|---|---|---|---|---|---|---|
| | Thresh - 1°C | Thresh. | Thresh + 1°C | Thresh - 1°C | Thresh. | Thresh + 1°C | Thresh - 1°C | Thresh. | Thresh + 1°C |
| HJA-CEN | 414.2 | 528.9 | 611.8 | 128 | 142 | 144 | 144 | 159 | 172 |
| HJA-VAN | 564.3 | 645.3 | 726.6 | 132 | 134 | 145 | 164 | 173 | 184 |
| HJA-UPL | 984.2 | 1165.5 | 1387.3 | 160 | 166 | 173 | 190 | 202 | 210 |
| SSC-LWR | 401.5 | 535.6 | 624.6 | 154 | 161 | 162 | 137 | 146 | 151 |
| SSC-UPR | 508.4 | 585.4 | 649.7 | 154 | 155 | 155 | 142 | 147 | 151 |
| YOS-DAN | 668.8 | 677.8 | 678.8 | 169 | 169 | 170 | 206 | 209 | 209 |
| JD-125 | 77.2 | 89.6 | 99.9 | 116 | 116 | 116 | 75 | 83 | 97 |
| JD-124b | 180.2 | 191.7 | 203.8 | 124 | 126 | 127 | 122 | 131 | 134 |
| JD-124 | 72.3 | 81.5 | 87.3 | 128 | 115 | 115 | 78 | 81 | 92 |
| NWT-C1 | 400.1 | 406.7 | NA | 204 | 204 | NA | 224 | 229 | NA |
| NWT-SDL | 914 | 914.6 | NA | 225 | 225 | NA | 240 | 241 | NA |

For the final question, "How important is it to use dew point or wetbulb temperature at warm sites versus at cold sites?", we would argue the best practice is to use a humidity-based temperature metric at all sites. Such methods better represent precipitation and

produce better model outcomes (e.g., Ding et al., 2014; Harder and Pomeroy, 2013, 2014; Harpold et al., 2017; Jennings et al., 2018; Marks et al., 2013). The bivariate binary logistic regression model, which performed best relative to other methods when compared to precipitation phase observations in a previous study (Jennings et al., 2018), produced snow cover metrics similar to the optimized threshold at most stations. It produced mean peak SWE, peak SWE DOWY, and SCD biases (relative to the optimized threshold) of -18.0 mm, 0.5 d, and -1.9 d, respectively.

Please note, we have not added the above material to the manuscript yet because it is consistent with the findings already presented in the submitted document. If you find this material worthy of inclusion, please let us know and we can add it as either supplementary material or as an appendix.

This having said, the study is nevertheless worth publishing and interesting for the readers of HESS. Below some general and detail comments.

**General comments**

I would not say that a study tests 12 different methods if only a few methods are tested with different parameter values; this oversells the study in the abstract. I would in fact say that the study tested four different methods: based on air temperature (with different 50% thresholds and different transition ranges, some of the ranges being 0), based on dew point and wet bulb temperature and based on binary regression.

Fair point. We have updated the text (see response to detailed comments below) to say we tested 5 different methods (counting the range as a different method than the threshold because the former produces mixed precipitation and the latter does not).

A key analysis of the paper is the one of "Climatic controls on precipitation phase method sensitivity".(section 4.4); it analyzes how the results vary with air temperature. Air temperature sensitivity is, however, built into each method in a different way. In the case of daily snowfall fraction: the fact that it shows the highest standard deviation for air temperatures between 0 and 4 C simply expresses the fact that several methods use thresholds in this range. The result would look different if the thresholds were between -2 and 2 C. This should be better reflected in the discussion of of the results.

Correct, the variability is tied to the methods themselves. However, we think it is important to present this information because the methods we used are based on empirical relationships (air temperature thresholds and ranges, dew point temperature thresholds), physical principles (wet bulb threshold to approximate hydrometeor

temperature (Harder and Pomeroy, 2013)), and statistical relationships (the binary logistic regression models). A threshold of -2°C would likely widen the range of variability but it would have no empirical, physical, or statistical relationship to precipitation phase partitioning except in some extremely rare, unique cases. Furthermore, this comment was similar to the feedback from Dr. Jono Conway, who noted the range in variability was likely produced by the extreme air temperature thresholds and ranges ($T_{a0}$, $T_{ar0}$, and $Ta_3$). To respond to his comment, we removed these methods and re-performed the analysis and the finding was the same (please see Figure R2 in our response to Dr. Conway). Even limiting the analysis to the most representative methods, the variability stays highest between 0°C and 4°C.

Additionally, it is essential to point out this air temperature range of variability for two reasons:

1. Areas most "at risk" to the snow-rain transition due to climate warming have seasonal air temperatures near and slightly above freezing (e.g., Nolin and Daly, 2006)
2. 0°C to 4°C is also the air temperature range where precipitation phase methods perform the worst (Ding et al., 2014; Jennings et al., 2018).

Thus, we have a compounded problem in that we are concerned with snow-to-rain shifts in areas with seasonal air temperatures where precipitation phase partitioning is most uncertain and our available methods exhibit downgraded performance. Given that we showed these areas (i.e., winter and spring air temperatures above freezing) also express the greatest sensitivity in terms of peak SWE magnitude and timing, plus snow cover duration, we think it is necessary to include this information.

In general, the conclusion that precipitation falling in the range 0 – 4 C explains much of the variation observed across the methods comes from the choice of the threshold values. Without actual comparison to observed data, the results are hard to generalize. Why is there no comparison to actual SWE-derived thresholds?

Please see our responses above.

Furthermore, when reading the results section where actual SWE curves are presented for the first time, it is a little disappointing to see that all studied sites show a typical seasonal snow cover with significant accumulation over many weeks. The most sen- sitive sites would typically be the ones where the snow cover might build up several times during the winter.

We should note here that the SWE curves as presented are daily averages (Fig. 4 in submitted manuscript), which has the affect of obscuring transience. As we mentioned in the Study sites and data section, the HJA and JD stations are sometimes transient (p. 4 lines 9-10 through p. 5 line 1, and p.5 lines 21-22) and they are most sensitive to phase method choice in terms of peak SWE magnitude (HJA only) as well as peak SWE timing and SCD (Fig. 5 in submitted manuscript).

**Detailed comments**

• The abstract does not mentioned what types of methods have been tested nor whether they have been compared to reference data or which method performed best

Yes, that is an oversight on our part. We have changed the abstract to note:

*"The methods in this study included different permutations of air, wet bulb, and dew point temperature thresholds, air temperature ranges, and binary logistic regression models."*

We have also added a line saying:

*"Compared to observations of snow depth and SWE, the binary logistic regression models produced the lowest mean biases, while high and low air temperature thresholds tended to respectively overpredict and underpredict snow accumulation."*

• Introduction: it would have been interesting to shortly discuss how /where pre-cipitation phase is actually observed; as far as I am aware of, actual precipitation phase observations are crucially missing at most places.

Good point. We have added a line to the first paragraph of the Introduction:

*"Complicating matters is the fact precipitation phase is rarely observed in mountain regions on a continuous bases over long time scales."*

• Introduction: the manuscript focuses its discussion on snow-hydrological models. How do meteorological forecast models determine the limit (elevation) of snow fall? Completing the literature review with this respect would complete the picture

This is covered in discussion (p. 21 lines 4-17) and not necessary for the Introduction as we do not utilize any atmospheric model methods in this work.

- P. 2: "In general, warmer sites are more sensitive to precipitation phase method selection in terms of annual snowfall fraction variability, though it is less certain how this variability translates into divergences in simulated snow accumulation and melt. " This statement is given without reference. In what is the apparently previously known result different from your own findings?

*Text changed to: "This previous work has shown, in general, warmer sites are more sensitive…" in order to clearly connect the statement with the published literature in the previous line.*

- Study sites: It might be useful to know the variability of the daily air temperature around the seasonal mean (ie. the anomalies, obtained e.g. by fitting a sine curve to air temperature as in the work of Woods, 2009. It is this variability that will tell something about the probability of switching from accumulation to melting conditions and about a site sensivitiy to the chosen temperature threshold.

*This sounds similar to the point raised by Nayak et al. (2010), who showed the effects of switching from sub-freezing to freeze-thaw diurnal cycles on snowpacks at Reynolds Creek. It is clear fluctuations above and below freezing having important effects on snow cover energetics. However, we are unclear as to what new, relevant information such data would provide to the current study. Perhaps we are misunderstanding the comment, so please clarify if so.*

- Methods: it is not clear at this stage that all stations always show a seasonal snow cover (significant accumulation over several weeks), which is important for the concept of "peak SWE" to be meaningful

*It is noted in the Study sites and data section (p. 4-5) for each location whether seasonal snowpacks develop or not.*

- the current definition of snowmelt rate is probably over sensitive to spurious shifts from a primary to a secondary SWE peak, which could reduce the melt dura- tion sensibly; how could this measure be made more robust? Similar comment applies to the peak SWE date that is discussed in the results section. Is this measure useful? Minor modifications of SWE accumulation can switch the SWE peak date between a spurious primary or secondary peak (Figure 4 suggest that stations with two peaks might exist, but I might be mistaken).

*We noted on p. 15 (lines 3-6): "We found the greatest differences in peak SWE dates*

*were generally simulated on years with low/transient snow cover. In these cases, late-season precipitation was simulated as rain by the low $T_a$ thresholds and snow by the high $T_a$ thresholds, meaning an early SWE maximum was recorded as the peak in the former case and a late SWE maximum in the latter case."* Given that peak SWE timing is an important measure of melt onset in the western US, we find it is necessary to highlight the variability in this metric as produced by different phase methods. Our finding indicates research on future changes to snowmelt timing is affected by modeling decisions on assigning precipitation phase.

Regarding snowmelt rate, we present the seasonal melt rate or ablation slope (e.g., Trujillo and Molotch, 2014) because of the importance of the spring snowmelt freshet to streamflow generation in many mountainous areas of the western US. However, we admit this overlooks the important winter contributions of snowmelt to groundwater and streamflow in maritime and transient snow environments. Switching the analysis to include all days when snowmelt was > 0 mm, we found marginal differences across the precipitation phase methods (mean differences were all less than 2.2 mm d$^{-1}$, which is less than the nominal precision of the SNOTEL snow pillows in the western US). Looking at daily average melt rate differences between the $T_{a0}$ and $T_{a3}$ thresholds helps illustrate why. Figure R3 below shows that generally $T_{a0}$ produces higher melt rates than $T_{a3}$ early in the snow cover season, while the reverse is true later in the season. Although annual average melt rates exhibit few differences, this figure shows the timing of terrestrial water inputs is important.

[Figure]

**Figure R3. The difference in daily average snowmelt rate between $T_{a3}$ and $T_{a0}$.**

- P. 14 "meaning a significant proportion of water was simulated to have run off using one precipitation phase method versus being stored in the snowpack". This not well formulated since rainfall does not necessarily run off. It can infiltrate and recharge the groundwater.

We agree this was imprecise wording. We have changed this to, *"meaning a significant proportion of water was simulated to have infiltrated or run off using one precipitation phase method versus being stored in the snowpack..."*

- Section 4.4: Here, standard deviations are calculated across the results of all 12 computation methods. Standard deviation does not seem to be a good measure to quantify the variability of values that do not come from an actual sample of a given process but of values pertaining to different methods. (Besides: how are standard deviations obtained? First per method and then averaged over all methods?)

The standard deviation values presented in Section 4.4 and Figure 6 are computed per air temperature and RH bin across all stations and methods as noted in the text. Although standard deviation is an appropriate metric of variability in this context, we redid the

analysis using the uncertainty formulation from Harder and Pomeroy (2014). We modified it to be per RH and temperature bin. The result was the same (Figure R4):

[Figure]

Figure R4. Same as Figure 6 in submitted manuscript, but standard deviation is replaced with the uncertainty metric from Eq. 1 in Harder and Pomeroy (2014).

Screenshot of text from Harder and Pomeroy (2014) showing the uncertainty metric equation:

$$uncertainty = \frac{\sum_{i=1}^{n}(\mathrm{Max}_i - \mathrm{Min}_i)}{n} \qquad (1)$$

where Min and Max refer to the lowest and highest values of a model output variable from the 63 model runs, $i$ is the index (time step) of the value and $n$ is the number of values (total time steps). The units of uncertainty are the same as the hydrological variable being considered. The uncertainty and differences between PPMs are summarized using mean values over an entire hydrological year (1 October–30 September).

**References**

Ding, B., Yang, K., Qin, J., Wang, L., Chen, Y. and He, X.: The dependence of precipitation types on surface elevation and meteorological conditions and its parameterization, J. Hydrol., 513, 154–163, 2014.

Harder, P. and Pomeroy, J.: Estimating precipitation phase using a psychrometric energy balance method, Hydrol. Process., 27(13), 1901–1914, doi:10.1002/hyp.9799, 2013.

Harder, P. and Pomeroy, J. W.: Hydrological model uncertainty due to precipitation-phase partitioning methods, Hydrol. Process., 28(14), 4311–4327, 2014.

Harpold, A. A., Kaplan, M., Klos, P. Z., Link, T., McNamara, J. P., Rajagopal, S., Schumer, R. and Steele, C. M.: Rain or snow: hydrologic processes, observations, prediction, and research needs, Hydrol Earth Syst Sci, 21, 1–22, 2017.

Jennings, K. S., Winchell, T. S., Livneh, B. and Molotch, N. P.: Spatial variation of the rain-snow temperature threshold across the Northern Hemisphere, Nat. Commun., 9, doi:10.1038/s41467-018-03629-7, 2018.

Marks, D., Winstral, A., Reba, M., Pomeroy, J. and Kumar, M.: An evaluation of methods for determining during-storm precipitation phase and the rain/snow transition elevation at the surface in a mountain basin, Adv. Water Resour., 55, 98–110, doi:10.1016/j.advwatres.2012.11.012, 2013.

Nayak, A., Marks, D., Chandler, D. G. and Seyfried, M.: Long-term snow, climate, and streamflow trends at the Reynolds Creek Experimental Watershed, Owyhee Mountains, Idaho, United States: CLIMATE TRENDS AT RCEW, Water Resour. Res., 46(6), n/a-n/a, doi:10.1029/2008WR007525, 2010.

Nolin, A. W. and Daly, C.: Mapping "at risk" snow in the Pacific Northwest, J. Hydrometeorol., 7(5), 1164–1171, 2006.

Rajagopal, S. and Harpold, A. A.: Testing and Improving Temperature Thresholds for Snow and Rain Prediction in the Western United States, JAWRA J. Am. Water Resour. Assoc. [online] Available from: http://onlinelibrary.wiley.com/doi/10.1111/1752-1688.12443/full (Accessed 23 August 2016), 2016.

Storck, P., Lettenmaier, D. P. and Bolton, S. M.: Measurement of snow interception and canopy effects on snow accumulation and melt in a mountainous maritime climate, Oregon, United States, Water Resour. Res., 38(11), 5–1, 2002.

Trujillo, E. and Molotch, N. P.: Snowpack regimes of the Western United States, Water Resour. Res., 50(7), 5611–5623, doi:10.1002/2013WR014753, 2014.

Wayand, N. E., Clark, M. P. and Lundquist, J. D.: Diagnosing snow accumulation errors in a rain-snow transitional environment with snow board observations, Hydrol. Process., 31(2), 349–363, doi:10.1002/hyp.11002, 2017.

Zhang, Z., Glaser, S., Bales, R., Conklin, M., Rice, R. and Marks, D.: Insights into mountain precipitation and snowpack from a basin-scale wireless-sensor network, Water Resour. Res., 53(8), 6626–6641, doi:10.1002/2016WR018825, 2017.

---

## Author Comment (AC2) · 31 May 2019

We thank Dr. Conway for their insightful review. Our responses can be found in blue throughout the following text. Please note, tables and figures specific to this response document are given with the prefix R (for example, Table R1 in the comment below). Tables and figures in the manuscript are referred to by numbers only.

Review of paper hess-2019-82 "**The sensitivity of modeled snow accumulation and melt to precipitation phase methods across a climatic gradient**" by Keith S. Jennings and Noah P. Molotch

Jono Conway

This paper presents a systematic evaluation of the impact of precipitation phase partitioning on modelled snowfall and snowpack evolution. Multi-year datasets from 11 stations across 5 locations in the western United States are used to drive simulations with a sophisticated snowpack physics model. The effect of parameter and algorithm choice are assessed for a range of commonly used parameterisations. The authors relate the modelled sensitivity to average climate characteristics. Snowfall and maximum accumulated snow in warmer maritime locations with high precipitation and winter temperatures between 0 and 4 degrees Celsius are found to be most sensitive to precipitation partitioning, while snowfall in colder inland locations are found to be less sensitive.

The manuscript is well written and with good figures and a clear systematic structure. It addresses a topic of high interest and relevance internationally. However, there are some areas that should be addressed before the paper could be accepted for publication.

**Major comments**

While the paper is framed as a comparison of methods used to partition precipitation, the results mainly reflect the range of Ta thresholds used (0 to 3 C) rather than the choice of parameterisation. This is in part due to the use the range metric on results that are generally are bounded by the two extreme Ta thresholds. The abstract and conclusions should reflect this (i.e. being explicit about choice of parameter values and/or parameterisation rather than using the ambiguous term "method". If the authors wish to make general statements, then using "precipitation partitioning" would be more appropriate. It is well established the Ta alone is a poor predictor of precipitation phase, so to really compare methods, those that perform poorly against observed SWE (e.g. Ta0 and Ta3) should be removed from the analysis. This would highlight the differences

induced by using different parameterisations that have a sound physical basis. If the Ta0 and Ta3 options are to be retained, then further justification for their inclusion should be given in the methods section. The dependence of the results (especially the range of Ta with a large range in modelled snow) on the range of Ta thresholds used should also be discussed. Perhaps the use of a standard deviation or similar metric rather than a range metric would put the focus on the choice of parameterisation. Further specific comments address this issue.

We should note here that at YOS-DAN, NWT-C1, and NWT-SDL, the $T_{d1}$ threshold produced greater annual snowfall fractions than $T_{a3}$. Thus, although $T_{a3}$ produced higher snowfall fractions at the remaining sites, this effect was by no means universal. Additionally, as mentioned in the Introduction (p. 3 lines 17-18) and further highlighted in our response to Dr. Schaefli, a 3°C $T_a$ threshold is appropriate in upland continental areas of the western US (e.g., NWT) where snowfall is more common at warmer temperatures than in other locales. Our concern is that many land surface and hydrologic models use spatially uniform air temperature thresholds to partition precipitation phase, so we argue that it is essential to incorporate thresholds that cover the range of observed rain-snow partitioning air temperatures for our study sites (1°C to 3°C). And, despite decades of evidence showing its inefficacy, the $T_{a0}$ threshold and $T_{ar0}$ range are still commonly employed to partition precipitation phase. For example, the widely used VIC macroscale hydrologic model assigns precipitation phase with a default -0.5°C to +0.5°C temperature range, centered on 0°C (https://github.com/UW-Hydro/VIC/; accessed 2019-05-20). We therefore included this method, if only to provide more evidence that it underpredicts snowfall, snow accumulation, and snow cover duration.

In order to further address the need to use the whole range of air temperature thresholds, we have added new text to the Methods (Sect. 3.2):

*"$T_a$ thresholds were chosen to represent the spatial variability of rain-snow partitioning in the western United States, where values of approximately 1°C are common near the Pacific Coast, increasing towards 3°C in the Rocky Mountains (Jennings et al., 2018). Additionally, despite significant literature showing its poor performance (e.g., Jennings et al., 2018; Marks et al., 2013), we included a 0°C $T_a$ threshold in the analysis because it is still widely used in observational and model-based hydrologic studies."*

In an early draft of this manuscript, we analyzed standard deviations in addition to the ranges presented in the submitted version. The story remained the same: warm maritime

sites were greatly impacted by precipitation phase method choice, while cold sites were not. This is illustrated in Figure R1 below:

[Figure]

**Figure R1. The annual standard deviation (left column) and range (right) in simulated peak SWE (ai,aii), peak SWE date (bi,bii), snow-off date (ci,cii), snow cover duration (di,dii), and melt rate (ei,eii) due to precipitation phase method selection at the study stations.**

Regarding the semantics of "method" versus "precipitation partitioning" versus "parameterization," "precipitation phase method" is commonly used to describe modeling and empirical approaches to discriminating between rain and snow (e.g., Harder and Pomeroy, 2014; Harpold et al., 2017). We will leave as is.

Timing and magnitude of SWE ranges seem mainly related to snowfall and accumulation, whereas as range of melt rate does not have high sensitivity or clear relation to climate. This should be clearer in the abstract and conclusions.

We added to text to the abstract noting this finding:

*"Average ranges in snowmelt rate were typically less than 4 mm d$^{-1}$ and exhibited little relationship to seasonal climate."*

And to the conclusion:

*"In contrast to the marked differences in peak SWE, melt onset, and snow cover duration between the warm and cold stations, ranges in snowmelt rate exhibited little relationship to seasonal climate."*

The abstract and conclusions need to highlight the novel aspects of the results presented here and provide clearer recommendations for future research. While the analysis is comprehensive, the result is not entirely new and, in my opinion, there are other results in the paper that could (and should) be highlighted in addition to the main result that the relative differences are largest in maritime snowpack. For example, the fact that using threshold or ranges for Ta (for the same 50% crossover) do not produce large differences in the snowpack, or that partitioning choice has little effect on snowmelt rate and the effects are dominated by snowfall. At present, the authors recommendations for future researchers are unclear.

For the ranges, we added text to Discussion Sect. 5.1:

*"In the course of this work we found negligible differences between $T_{a0}$ and $T_{r0}$ as well as between $T_{a1}$ and $T_{r1}$ in terms of annual snowfall fraction (Fig. 5) and model performance (Fig. 3). This suggests the ranges and the mixed-phase precipitation they produced provided little further information on precipitation phase at the hourly model time scale relative to the thresholds. However, it should be noted there is relatively little*

*quantitative data on the frequency and solid-liquid proportions of mixed-phase events (e.g., Yuter et al., 2006). Work from the Torino region of Italy showed mixed-phase events are relatively few compared to all-rain and all-snow events (Avanzi et al., 2014), while research in a maritime climate indicated mixed-phase events can be quite frequent (Wayand et al., 2016). Thus, future work would benefit from further explorations of the frequency of mixed-phase events and model representations thereof at multiple time scales."*

For snowmelt rate, it is not that the effect is small (Table 5 shows relative differences between 11.5% and 235.5%), it is that the metric showed no relationship to seasonal climate. Please see our response to the comment above for the extra material we added on snowmelt rate.

For novelty/implications, we changed the final line of the abstract to:

*"This study shows care should be taken when selecting a precipitation phase method as the variability introduced to snow accumulation and melt will likely propagate into simulated streamflow and land surface albedo, particularly at the warmer fringes of the seasonal snow zone."*

Regarding future directions, suggestions were given in the original manuscript (p. 8 lines 15-16, p. 20 lines 25-27, p. 21 lines 1-3, p. 22 lines 16-19, p. 23 lines 13-16). Given the multiple lines devoted to this topic and the further additions noted in this response, we find no further recommendations are needed.

The use of multiple linear regression is probably not appropriate here, but if retained should be presented and discussed more fully.

Please see our response to the specific comment below on this topic.

**Specific comments (page-line)**

1-15 please be clear the study modelled non-vegetated snowpacks only.

We have reconfigured the abstract to note these were point simulations with no canopy cover.

4-3 Given that they form a key part of the results, please include average values for $T_w$ and $T_d$ in Table 1.

Added to Table 1

7-26 The large bias in LWin is concerning – perhaps the influence of vegetation on the measurements whereas LWin is modelled for non-vegetated location? This should be discussed when presenting the validation results in Figure A2.

We added more text expanding upon the bias in Sect. 3.1:

"*At the HJA stations, we bias-corrected the $LW_{in}$ estimate based on one year of $LW_{in}$ observations from HJA-VAN that showed a -56.9 $W\ m^{-2}$ wintertime bias, which may have been related to site vegetation conditions. This was significantly larger in magnitude than the bias found in the Unsworth and Monteith (1975) estimate by Flerchinger et al. (2009), suggesting its performance is more spatially variable than previously noted. This finding also underscores the need for enhanced monitoring of the radiation budget at snow modeling sites (Lapo et al., 2015; Raleigh et al., 2015, 2016).*"

17-8 "80.1% of the variance in annual snowfall fraction standard deviation" – the figure caption and methods describes this as the "range in annual snowfall fraction" – please clarify which it is and correct.

Yes, good catch. We have changed the text to "…*annual snowfall fraction range*" to match the figure.

18-1 Figure 6 and 7 – given that the range in snowfall fraction is driven primarily by the two extreme air temperature threshold methods (Ta0 and Ta3) these results are presumably quite sensitive to the choice of the Ta thresholds? Please discuss and if possible show the sensitivity of the results to the choice of threshold.

Yes, Figure 6 is designed to illustrate the effect of threshold/method choice on daily snowfall fraction. The data presented are standard deviations, which minimizes the effect of the extreme $T_a$ thresholds. However, we were curious how removing $T_{a0}$, $T_{ar0}$, and $T_{a3}$ would affect the analysis and we found it made little difference (Figure R2 is nearly identical to Figure 6 in the submitted manuscript). Except for 1 outlier at -19.5% all differences in the standard deviations for the $T_a$ and RH bins are between -7% and +5%, with a mean difference of -1.3% (computed by subtracting the SD for the analysis with all methods included from the analysis with $T_{a0}$, $T_{ar0}$, and $T_{a3}$ removed).

[Figure]

**Figure R2. Same as Figure 6 in submitted manuscript but with $T_{a0}$, $T_{ar0}$, and $T_{a3}$ removed from the analysis.**

Given this analysis held up to the removal of the three least physically representative thresholds, we find the inclusions of Figures 6 and 7 along with the associated text to be appropriate.

18-6 Looking at the figure, it seems that a multiple linear regression may not be appropriate. There seems to be two groupings – highly sensitive warm and wet locations, less sensitive drier locations that span both warm and cold locations. Also, given that the equation is not presented nor used further, and the issues discussed with extrapolating the equations, please consider removing the regression. If it is retained, please present the equation and display contours of predicted values on Figure 8 so that the reader can visualise the predicted relationships.

We have decided use a loess function to create a smooth surface presented behind the station data (new Fig. 10 shown below). This, we believe, more clearly shows the clustering of low peak SWE ranges at the colder and/or low precipitation sites and high peak SWE ranges at the maritime sites without introducing the statistical pitfalls of multiple linear regression. We also edited the text to remove the linear regression statements.

*"We next evaluated how sensitivity in peak SWE was related to seasonal climate. In this case, warmer $T_a$ and increased PPT were both associated with greater ranges in the peak SWE simulated by the different precipitation phase methods (Fig. 10). This meant the maritime sites HJA and SSC had the greatest sensitivity to precipitation phase method due to their relatively warm $T_a$ and high PPT values. Conversely, moderate PPT values and lower $T_a$ led to minimal sensitivity at the cold continental NWT stations and the cold maritime YOS-DAN station. Again, the effect of $T_a$ on sensitivity was manifest in the data. In high snowfall years at NWT-SDL, Dec–May PPT approached that of the low Dec–May PPT years at HJA and SSC. However, despite the increased PPT at NWT-SDL, the range in peak SWE predicted by the different precipitation phase methods remained low."*

[Figure]

*Figure 10. Range in annual peak SWE as simulated by the different precipitation phase methods at the 11 study stations. Each point represents one simulation year at a given station and larger points correspond to larger differences in maximum minus minimum peak SWE. The background shading corresponds to ranges in peak SWE predicted by a loess function fit to the station data.* **[Please note, this figure has changed from 8 in the submitted manuscript to 10 in the revised version because we moved the validation figures from the appendix to the results as per the recommendation below.]**

19-10 The validation results presented in the appendix should be included in the results or methods section, especially as they form part of the discussion, rather than simply an

intermediate methodological step.

We have moved this from the Appendix to be the first results section:

We have also added material in the Methods detailing how validation was performed (appended to the end of Sect. 3.1):

*"To validate model output, we compared simulated SWE and snow depth to observations at our study stations. SWE was observed at all HJA stations, SSC-UPR, YOS-DAN, and both NWT stations, while snow depth was observed at all HJA stations, both SSC stations, and all JD stations. All SWE data were derived from automated snow pillow measurements except for NWT-SDL, which was acquired through manual snow pit observations (Williams, 2016). Similarly, automated ultrasonic snow depth sensors produced all snow depth data. Comparisons were made at the daily time scale when either simulated or observed SWE or snow depth were > 0 mm. This was done to prevent artificial enhancement of objective function values during periods when snow cover was absent."*

We have also edited discussion Sect. 5.1 to reflect these changes and to incorporate feedback from the two subsequent comments.

[revised manuscript text omitted]

*precipitation phase method across a climatic gradient. We did not create optimized model setups at each site, but rather kept model setup consistent in order to compare the sensitivity of phase partitioning without introducing other uncertainties. Thus, the low $r^2$ and higher bias values at HJA-VAN, NWT-SDL, and JD-124 (Fig. 4) could likely be improved with model tuning, but we did not pursue such an approach."*

20-3 "In that context, one can consider the RegBi model as a baseline given its top rank in a Northern Hemisphere precipitation phase method comparison". Please describe and discuss the results presented here (figure A1) that seem to show similar performance for a range of methods that incorporate humidity information. The discussion as it is not balanced and does not accurately reflect the results presented. Please revise.

Please see edited discussion Sect. 5.1 above.

20-11 "a referendum." This does not seem an appropriate term – please revise. You could either give an expert view based on the results presented here, or cite others work.

Please see edited discussion Sect. 5.1 above.

20-24 "Therefore, our use of a single model may overestimate or underestimate the sensitivity of snow cover evolution to precipitation phase method at certain sites and points in time." This statement is very broad - more effort is needed to quantify and discuss the uncertainty of the model simulations.

This statement is broad because model intercomparisons say little about the effect of precipitation phase method selection. For example, SnowMIP2 used different precipitation phase methods at different sites (Rutter et al., 2009). Thus it is still unknown how model selection and phase partitioning methods interact (i.e., would a temperature index model be more affected than a physics-based model?). We stand by our statement but have clarified with some extra text:

*"Given this variable performance and differences in snow model structure and physics, it is possible that some models may be more or less sensitive to the choice of a precipitation phase method. Our use of a single model may overestimate or underestimate the sensitivity of snow accumulation and melt to precipitation phase method selection. Future research should therefore focus on how model choice affects the sensitivity of simulated*

*snow cover evolution to precipitation phase method."*

21-23 "These large variations in snow cover evolution were likely due to the combined effect of reduced frozen mass entering the snowpack and subsequent changes to the snowpack energy balance". More detailed results are needed to support this statement. For example, the change in snowfall mass and albedo could be shown to illustrate the importance of the direct and indirect effects on snowpack mass balance.

There are 10 citations in the previous lines detailing how rain vs. snow affects the snowpack energy balance. We include this as a discussion because a full treatment of the energy balance data is outside of the scope of this already fairly long manuscript.

22-3 "In this context, the precipitation phase methods that produced more rainfall affected snow cover evolution not just through reduced frozen mass but also through changes to the snowpack energy budget." These results are not shown here (they could be?) so this statement is speculation. Please revise.

See response to comment above. We also added a qualifier to the sentence and followed it with a future research line:

*"In this context, the precipitation phase methods that produced more rainfall likely affected snow cover evolution not just through reduced frozen mass but also through changes to the snowpack energy budget. Further observational and modeling research is warranted to evaluate how rain versus snow affects snowpack energetics."*

22-25 "winter and spring average $T_a$ values (0°C–4°C) that lead to the greatest uncertainty in rain- snow partitioning," I would argue that the uncertainty is not in the actual rain-snow partitioning, but rather due to the use of an inappropriate parameterisation (only $T_a$) which requires a wide range of parameter tuning. Please revise.

As we showed in our response to a comment above, the range of uncertainty stays the same when removing $T_{a0}$, $T_{ar0}$, and $T_{a3}$. Furthermore, the difficulty of predicting precipitation phase and the resulting uncertainty at temperatures slightly above freezing is a well known phenomenon in both hydrology (Ding et al., 2014; Harpold et al., 2017; Jennings et al., 2018) and atmospheric science (Ralph et al., 2005; Stewart et al., 2015).

23-1 Please mention that no clear relationship was found for snowmelt rate in the

conclusions – this is still a key result and an important caveat to the earlier statement that "precipitation phase method introduced significant variability into simulated snow accumulation and melt".

Added per recommendation on an earlier comment.

23-30 How was the r2 calculated here? the average r2 of hourly SWE/snowdepth or something else? Please include in the text and figure caption.

Added to Methods Sect. 3.1 as noted above.

24-1 Given the poor performance of some methods (Ta0 ,Ta3, Tr0) should they be excluded from the analysis? If not, further discussion is needed.

Please see earlier comments on the $T_{a0}$, $T_{ar0}$ and $T_{a3}$ methods.

24-6 "at all stations." Given that SWE and snowdepth are only presented for some sites in Figure A2, I presume not all sites contribute to averages here? Please list the sites that contribute to each of the SWE and snowdepth validation statistics in the text or caption.

Please see new results text above.

Figure A2 – why is the snowdepth bias 0 for the JD sites?

That is an artifact of the data at JD. The low overall snow depth produced low absolute biases. We added this text to the figure caption (please note, this figure and section has been moved to Results 4.1 per recommendation on an earlier comment):

*"Note: in panel (c) the low mean biases for JD snow depth are due to small observed snow depth values at the site. Mean relative biases at these stations were 35.4% (JD-125), 3.8% (JD-124b), and 35.7% (JD-124)."*

**Editorial comments:**

10-6 "daily Ta and RH" do you mean "daily average Ta and RH"?

Yes, line changed to "daily average $T_a$ and RH"

16-5 "not computed because for" -> "not computed for"

Redundant "because" has been removed.

---

## Author Response (AR1)

Dear Dr. Markus Hrachowitz,

Please find included in this file a list of the major changes made to the manuscript along with our responses to the reviews by Drs. Schaefli and Conway and a tracked changes version of the revised manuscript.

Per your recommendations on major revisions we have substantially increased the text and analysis devoted to the observational SWE and snow depth data as well as adding material on transient snow.

We are thankful for the time invested by you and the reviewers. We feel your suggestions have substantially improved the revised manuscript.

Best regards, Keith Jennings and Noah Molotch 2019-07-16 Changes made to manuscript (for minor text edits as well as figure and section renumbering, please see tracked changes manuscript):

- 1. Updated abstract text per reviewer recommendations.
- 2. Updated introduction text per reviewer recommendations.
- 3. Added wet bulb and dew point temperature data to Table 1 per Dr. Conway's recommendation.
- 4. Added more information on longwave radiation and model validation to Sect. 3.1 per reviewer recommendations.
- 5. Added motivation for using the different Ta thresholds per reviewer recommendations.
- 6. Added section 3.4 on identifying and deviating from an optimized threshold per Dr. Schaefli's recommendation.
- 7. Removed multiple linear regression from Sect. 3.5 per Dr. Conway's recommendation.
- 8. Moved model validation to Sect. 4.1 from Appendix and updated the text and figures. Note: the original snow depth validation performed at JD stations was done in cm. This was changed to mm to be consistent with the other sites.
- 9. Added new paragraph to Sect. 4.4 and that includes further text on SWE/depth observations along with how transient snow is particularly affected by method choice in terms of snow cover duration. Also added Fig. 8 to display this information visually.
- 10. Added Sect. 4.5 and Fig. 9 per Dr. Schaefli's recommendation.
- 11. Changed Fig. 12 to remove multiple linear regression and add loess surface.
- 12. Rewrote Sect. 5.1 and updated other discussion sections per review recommendations.
- 13. Updated conclusions per reviewer recommendations.
- 14. Added model, code, and plot color ramp info to "Code and data availability" section.
- 15. Added "Competing interests" section.
- 16. Added Table S2 and Figs. S12 and S13 to the supplement per Dr. Schaefli's recommendation. Note: Figs. S12 and S13 are different from R1 and R2 in the response to Dr. Schaefli because our original analysis included snow-free periods. These were removed and the snowfall frequency curves were recalculated.

We thank Dr. Schaefli for their thoughtful review. Our responses can be found in blue throughout the following text. Please note, tables and figures specific to this response document are given with the prefix R (for example, Table R1 in the comment below). Tables and figures in the manuscript are referred to by numbers only.

This well written paper analyzes a key question for snow hydrology, which is the im- pact of precipitation phase algorithms on snow water equivalent (SWE) modelling in different climates. The paper studies four more or less different methods of precipita- tion phase computation (each with different portioning parameters) and assesses the impact of the methods on different snow accumulation and melt metrics, obtained with the model SNOWPACK at five different locations in the US. The methods are based on temperature thresholds and on bilinear regression. The analysis gives an answer to the general question of how important it is to carefully choose the precipitation phase method for different climates.

A drawback of the study is that it is purely simulation-based and does not use observed SWE data to push the study further. In fact, with the observed SWE data and SNOW- PACK, it might have been possible to estimate actual daily or hourly snow accumulation amounts and compute best parameter values for the studied precipitation phase meth- ods at the selected stations. This way, it would have been possible to judge how critical deviations from these best estimates would be at the different sites. In other words, this would allow to answer questions like "how critical is it to have a 1°C error in the air temperature threshold at a warm site as opposed to a cold site"? "How important is it to use dew point or wetbulb temperature at warm sites versus at cold sites?"

We agree this is a drawback of both this study and many other snow modeling research projects. There are, unfortunately, scant direct observations of precipitation phase in mountain regions. One of the few studies we are aware of that uses observations of precipitation phase—in this case snow board measurements—showed rain-snow partitioning errors can lead to significant biases in modeled snow accumulation at a site with a maritime climate similar to the HJ Andrews and Southern Sierra (Wayand et al., 2017). There is also evidence suggesting an optimized air temperature threshold varies throughout the snow season (Storck et al., 2002), meaning no single air temperature threshold (or range) would be applicable across sites and times.

At our study sites, there are no direct observations of precipitation phase, but we were interested in pursuing your question further. Table R1 below shows the optimized rain-snow air temperature threshold using four different data sources for each station. The second and third columns (Map and Obs.) correspond to data from earlier work that examined the spatial variability of rain-snow partitioning across the Northern Hemisphere (Jennings et al., 2018). The methods, quoted from the paper, are as follows:

"To construct a spatially continuous 50% rain–snow  $T_s$  [air temperature] threshold product across the Northern Hemisphere, we applied the optimized bivariate model to the MERRA-2 gridded reanalysis dataset63,64. Hourly 2 m  $T_s$ , specific humidity (q),  $P_s$ , and precipitation data were accessed from 1980 through 2007 and summarized to a daily time step. RH was calculated from the MERRA-2 data using an empirical equation as a function of q,  $P_s$ , and  $T_s$ . Daily snowfall probability was then simulated for each grid cell using the bivariate model when precipitation was greater than 1 mm and  $T_s$  fell within the range of -8 to 8 °C. We then calculated the 50% rain–snow  $T_s$  threshold by fitting the hyperbolic tangent to binned estimates of snowfall frequency per MERRA-2 grid cell using Eq. 1."

"We classified precipitation reports as either rain or snow using the World Meteorological Organization precipitation phase categories described in detail in  $Dai^{40,61}$ . Precipitation amounts were not included in the dataset and we removed sleet as well as potential mixed-phase observations from the analysis because the relative proportions of solid and liquid precipitation during such events were not reported (i.e., it was impossible to quantify the amount of precipitation falling as snow versus rain). The classification of precipitation events was then used to quantify the rain–snow frequency per 1 °C Ts bin from –8 to 8 °C at each station. In other words, if there were 100 total precipitation observations from 1 to 2 °C, 75 of which were snow, the snowfall frequency in that bin would be 75.0%. We then calculated the 50% rain–snow Ts threshold for each station using the approach of  $Dai^{40}$ , where a sigmoidal curve is fit to observations of snowfall frequency per 1 °C Ts bin from –8 to 8 °C using a hyperbolic tangent function:

$$T_{50} = \frac{\tanh^{-1}\left(\frac{F}{a}+d\right)}{b} + c \tag{1}$$

where  $T_{50}$  equals the 50% rain–snow  $T_s$  threshold (°C), F equals snowfall frequency (in this case 0.5, dimensionless), and a, b, c, and d are the fitting parameters (dimensionless)."

The fourth and fifth columns in Table R1 use changes in SWE and snow depth to estimate a rainsnow air temperature threshold. We used a modified version of the approach of Rajagopal and Harpold (2016) to predict precipitation phase by designating a daily increase of SWE or snow depth as snowfall and a zero change or decrease as rainfall when precipitation was greater than 2.54 mm and SWE or snow depth was greater than 0 mm. As with the Map and Obs. methods detailed above, we then binned snowfall frequency per 1°C air temperature bin (Figures R1 and R2) and computed the rain-snow air temperature threshold using Eq. 1 above. The SWE approach yielded values that approximated the Map and Obs. methods, but the depth-derived values were significantly lower. We would thus argue that this method was not appropriate for our purposes, although previous work has shown it to reasonably estimate precipitation phase at subdaily time scales (e.g., Marks et al., 2013; Zhang et al., 2017).

**Table R1.** Optimized rain-snow air temperature thresholds for each station in the study using four different data sources: 1-Map) The spatially continuous threshold map from Jennings et al. (2018) created using reanalysis data from MERRA-2 and the bivariate binary logistic regression model; 2-Obs.) The observed threshold from the closest meteorological station (Jennings et al., 2018); 3-SWE) The threshold inferred from changes in SWE at each study station (Fig. R1); 4-Depth) The threshold inferred from changes in snow depth at each study station (Fig. R2). An NA indicates there were insufficient data to estimate the threshold from SWE and/or snow depth.

|         | Optimi | zed rain-snow ai | r temperature th | reshold (°C) |
|---------|--------|------------------|------------------|--------------|
| Station | Map    | Obs.             | SWE              | Depth        |
| HJA-CEN | 1.19   | 1.12             | 1.29             | -0.24        |
| HJA-VAN | 1.19   | 1.12             | 0.8              | -0.84        |
| HJA-UPL | 1.19   | 1.12             | -0.4             | -0.81        |
| SSC-LWR | 1.7    | 1                | NA               | 0.14         |
| SSC-UPR | 1.7    | 1                | 0.87             | -0.34        |
| YOS-DAN | 2.21   | 2.78             | NA               | NA           |
| JD-125  | 2.25   | 1.25             | NA               | -0.97        |
| JD-124b | 2.25   | 1.25             | NA               | -1.91        |
| JD-124  | 2.25   | 1.25             | NA               | 0.41         |
| NWT-C1  | 2.84   | 2.34             | 3.57             | NA           |
| NWT-SDL | 2.84   | 2.34             | NA               | NA           |

Optimized rain-snow air temperature threshold (°C)

Figure R1. Snowfall frequency per 1°C air temperature bin as computed from SWE data. On days with precipitation > 2.54 mm, an increase in SWE was designated as a snowfall event, while a zero change or decrease in SWE was designated as rainfall.

Figure R2. Snowfall frequency per 1°C air temperature bin as computed from snow depth data. On days with precipitation > 2.54 mm, an increase in snow depth was designated as a snowfall event, while a zero change or decrease in snow depth was designated as rainfall.

Returning to the question of "how critical is it to have a 1°C error in the air temperature threshold at a warm site as opposed to a cold site," we analyzed the effect of deviating by 1°C from the mean threshold as calculated from the Map and Obs. columns in Table R1. In this context we rounded to the nearest integer degree to be consistent with our thresholds, giving the HJA stations a 1°C threshold, SSC a 1°C threshold, YOS a 2°C threshold, JD a 2°C threshold, and NWT a 3°C threshold. Because we did not include a 4°C air temperature threshold in our phase methods, we could only analyze a negative deviation at NWT. In Table R2 below, we present the mean peak SWE, peak SWE day of water year (DOWY), and snow cover duration (SCD) using the optimized air temperature threshold (center column, abbreviated Thresh.), the optimized threshold minus 1°C (left column, Thresh - 1°C), and the optimized threshold + 1°C (right column, Thresh + 1°C). Consistent with our findings in the paper, the warm maritime HJA and SSC stations are profoundly affected by deviations from the optimized threshold. Differences at these sites produced by deviating by  $\pm 1^{\circ}$ C from the optimized thresholds range between 141 and 403 mm for peak SWE, 1 and 16 d for peak SWE DOWY, and 9 and 29 d for SCD. Compare this to 1 to 10 mm for peak SWE, 0 to 1 d for peak SWE DOWY, and 1 to 5 d for SCD at the YOS and NWT stations. The consistent story is again that threshold choice makes a much larger impact at a warm site relative to a cold one.

|         | Mear   | peak SWE | (mm)   | Mean pe | eak SWE D | OWY (d) | Mean SCD (d) |         |        |  |
|---------|--------|----------|--------|---------|-----------|---------|--------------|---------|--------|--|
|         | Thresh |          | Thresh | Thresh  |           | Thresh  | Thresh       |         | Thresh |  |
| Station | - 1°C  | Thresh.  | + 1°C  | - 1°C   | Thresh.   | + 1°C   | - 1°C        | Thresh. | + 1°C  |  |
| HJA-CEN | 414.2  | 528.9    | 611.8  | 128     | 142       | 144     | 144          | 159     | 172    |  |
| HJA-VAN | 564.3  | 645.3    | 726.6  | 132     | 134       | 145     | 164          | 173     | 184    |  |
| HJA-UPL | 984.2  | 1165.5   | 1387.3 | 160     | 166       | 173     | 190          | 202     | 210    |  |
| SSC-LWR | 401.5  | 535.6    | 624.6  | 154     | 161       | 162     | 137          | 146     | 151    |  |
| SSC-UPR | 508.4  | 585.4    | 649.7  | 154     | 155       | 155     | 142          | 147     | 151    |  |
| YOS-DAN | 668.8  | 677.8    | 678.8  | 169     | 169       | 170     | 206          | 209     | 209    |  |
| JD-125  | 77.2   | 89.6     | 99.9   | 116     | 116       | 116     | 75           | 83      | 97     |  |
| JD-124b | 180.2  | 191.7    | 203.8  | 124     | 126       | 127     | 122          | 131     | 134    |  |
| JD-124  | 72.3   | 81.5     | 87.3   | 128     | 115       | 115     | 78           | 81      | 92     |  |
| NWT-C1  | 400.1  | 406.7    | NA     | 204     | 204       | NA      | 224          | 229     | NA     |  |
| NWT-SDL | 914    | 914.6    | NA     | 225     | 225       | NA      | 240          | 241     | NA     |  |

**Table R2.** Mean peak SWE, peak SWE DOWY, and SCD at the study stations using an optimized air temperature threshold as well as  $-1^{\circ}$ C and  $+1^{\circ}$ C deviations from the threshold.

For the final question, "How important is it to use dew point or wetbulb temperature at warm sites versus at cold sites?", we would argue the best practice is to use a humidity-based temperature metric at all sites. Such methods better represent precipitation and produce better model outcomes (e.g., Ding et al., 2014; Harder and Pomeroy, 2013, 2014; Harpold et al., 2017;

Jennings et al., 2018; Marks et al., 2013). The bivariate binary logistic regression model, which performed best relative to other methods when compared to precipitation phase observations in a previous study (Jennings et al., 2018), produced snow cover metrics similar to the optimized threshold at most stations. It produced mean peak SWE, peak SWE DOWY, and SCD biases (relative to the optimized threshold) of -18.0 mm, 0.5 d, and -1.9 d, respectively.

Please note, we have not added the above material to the manuscript yet because it is consistent with the findings already presented in the submitted document. If you find this material worthy of inclusion, please let us know and we can add it as either supplementary material or as an appendix.

This having said, the study is nevertheless worth publishing and interesting for the readers of HESS. Below some general and detail comments.

**General comments**

I would not say that a study tests 12 different methods if only a few methods are tested with different parameter values; this oversells the study in the abstract. I would in fact say that the study tested four different methods: based on air temperature (with different 50% thresholds and different transition ranges, some of the ranges being 0), based on dew point and wet bulb temperature and based on binary regression.

Fair point. We have updated the text (see response to detailed comments below) to say we tested 5 different methods (counting the range as a different method than the threshold because the former produces mixed precipitation and the latter does not).

A key analysis of the paper is the one of "Climatic controls on precipitation phase method sensitivity".(section 4.4); it analyzes how the results vary with air temperature. Air temperature sensitivity is, however, built into each method in a different way. In the case of daily snowfall fraction: the fact that it shows the highest standard deviation for air temperatures between 0 and 4 C simply expresses the fact that several methods use thresholds in this range. The result would look different if the thresholds were between -2 and 2 C. This should be better reflected in the discussion of of the results.

Correct, the variability is tied to the methods themselves. However, we think it is important to present this information because the methods we used are based on empirical relationships (air temperature thresholds and ranges, dew point temperature thresholds), physical principles (wet bulb threshold to approximate hydrometeor temperature (Harder and Pomeroy, 2013)), and statistical relationships (the binary logistic regression models). A threshold of -2°C would likely widen the range of variability but it would have no empirical, physical, or statistical relationship

to precipitation phase partitioning except in some extremely rare, unique cases. Furthermore, this comment was similar to the feedback from Dr. Jono Conway, who noted the range in variability was likely produced by the extreme air temperature thresholds and ranges ( $T_{a0}$ ,  $T_{ar0}$ , and  $Ta_3$ ). To respond to his comment, we removed these methods and re-performed the analysis and the finding was the same (please see Figure R2 in our response to Dr. Conway). Even limiting the analysis to the most representative methods, the variability stays highest between 0°C and 4°C.

Additionally, it is essential to point out this air temperature range of variability for two reasons:

- 1. Areas most "at risk" to the snow-rain transition due to climate warming have seasonal air temperatures near and slightly above freezing (e.g., Nolin and Daly, 2006)
- 2. 0°C to 4°C is also the air temperature range where precipitation phase methods perform the worst (Ding et al., 2014; Jennings et al., 2018).

Thus, we have a compounded problem in that we are concerned with snow-to-rain shifts in areas with seasonal air temperatures where precipitation phase partitioning is most uncertain and our available methods exhibit downgraded performance. Given that we showed these areas (i.e., winter and spring air temperatures above freezing) also express the greatest sensitivity in terms of peak SWE magnitude and timing, plus snow cover duration, we think it is necessary to include this information.

In general, the conclusion that precipitation falling in the range 0 - 4 C explains much of the variation observed across the methods comes from the choice of the threshold values. Without actual comparison to observed data, the results are hard to generalize. Why is there no comparison to actual SWE-derived thresholds?

**Please see our responses above.**

Furthermore, when reading the results section where actual SWE curves are presented for the first time, it is a little disappointing to see that all studied sites show a typical seasonal snow cover with significant accumulation over many weeks. The most sen- sitive sites would typically be the ones where the snow cover might build up several times during the winter.

We should note here that the SWE curves as presented are daily averages (Fig. 4 in submitted manuscript), which has the affect of obscuring transience. As we mentioned in the Study sites and data section, the HJA and JD stations are sometimes transient (p. 4 lines 9-10 through p. 5 line 1, and p.5 lines 21-22) and they are most sensitive to phase method choice in terms of peak SWE magnitude (HJA only) as well as peak SWE timing and SCD (Fig. 5 in submitted manuscript).

**Detailed comments**

• The abstract does not mentioned what types of methods have been tested nor whether they have been compared to reference data or which method performed best

Yes, that is an oversight on our part. We have changed the abstract to note:

"The methods in this study included different permutations of air, wet bulb, and dew point temperature thresholds, air temperature ranges, and binary logistic regression models."

We have also added a line saying:

"Compared to observations of snow depth and SWE, the binary logistic regression models produced the lowest mean biases, while high and low air temperature thresholds tended to respectively overpredict and underpredict snow accumulation."

• Introduction: it would have been interesting to shortly discuss how /where pre- cipitation phase is actually observed; as far as I am aware of, actual precipitation phase observations are crucially missing at most places.

Good point. We have added a line to the first paragraph of the Introduction:

"Complicating matters is the fact precipitation phase is rarely observed in mountain regions on a continuous bases over long time scales."

• Introduction: the